# SMAGA: Secondary Motion-Aware 3D Clothed Gaussian Avatars from Monocular Videos

**Seungeun Lee[1], SeungJun Moon[2], Hah Min Lew[3], Ji-Su Kang[1], Gyeong-Moon Park[3]***

[1]Klleon AI Research, [2] RLWRLD, [3]Korea University

## Abstract

Recent advances in neural rendering, particularly 3D Gaussian Splatting (3DGS), have enabled animatable 3D human avatars from single videos with efficient rendering and high fidelity. However, current methods struggle with dynamic appearances, especially in loose garments (e.g., skirts), causing unrealistic cloth motion and needle artifacts. This paper introduces a novel approach to dynamic appearance modeling for 3DGS-based avatars, focusing on loose clothing. We identify two key challenges: (1) limited Gaussian deformation under pre-defined template articulation, and (2) a mismatch between body-template assumptions and the geometry of loose apparel. To address these issues, we propose a motion-aware autoregressive structural deformation framework for Gaussians. We structure Gaussians into an approximate graph and recursively predict structure-preserving updates, yielding realistic, template-free cloth dynamics. Our framework enables robust dynamic appearance modeling under the single-view constraint, producing accurate foreground silhouettes and precise alignment of Gaussian points with clothed shapes. To demonstrate the effectiveness of our method, we introduce an evaluation dataset featuring subjects performing dynamic movements in loose clothing, and extensive experiments validate that our approach significantly outperforms existing 3DGS-based methods in modeling dynamic appearances from monocular videos.

## 1 Introduction

Creating animatable 3D avatars from monocular videos involves reconstructing a lifelike, controllable representation of a person capable of replicating both primary motions (i.e., movements of major body parts), and secondary motions (i.e., time-varying cloth dynamics). Achieving this capability is critical for immersive experiences in fields such as virtual reality, telepresence, and interactive entertainment, where realistic human representations significantly enhance user engagement (Sutherland et al., 1965; Lee et al., 2024). With the advent of 3D Gaussian Splatting (3DGS) (Kerbl & et al., 2023), high-quality neural rendering becomes feasible, substantially improving avatar realism and efficiency in synthesis from monocular videos.

However, existing 3DGS-based avatar methods (Lei et al., 2024; Hu et al., 2024a; Moon et al., 2024; Qian et al., 2024b) predominantly excel at modeling primary motion but exhibit limitations in robustly capturing secondary motion, as illustrated in Fig. 1a, where it shows the animation result of the 3D Gaussian avatar in a novel pose exhibiting a dynamic posture unseen during training. This is due to the fact that they rely on skeletal skinning of coarse meshes for animation (Loper et al., 2015), which inherently lacks subtle deformation effects such as inertia-driven soft-tissue dynamics. Consequently, it remains challenging to consistently reproduce these nuanced motions using neural networks alone.

There are two major challenges that hinder the secondary motion-aware dynamic appearance modeling in creating an animatable 3DGS avatar from a single video: (1) temporal context-unaware Gaussian deformation, and (2) cloth shape-agnostic Gaussian point initialization. First, existing methods deform the Gaussians of a clothed avatar as a function of the current body pose, which is aligned with the sampled images, to learn the appearance of a 3D Gaussian avatar. While this approach effectively captures the primary motion of main body parts, it is limited in representing secondary motion in garments such as dresses and skirts, which is strongly entangled with temporal continuity (Fig. 1b).

---

*Corresponding author

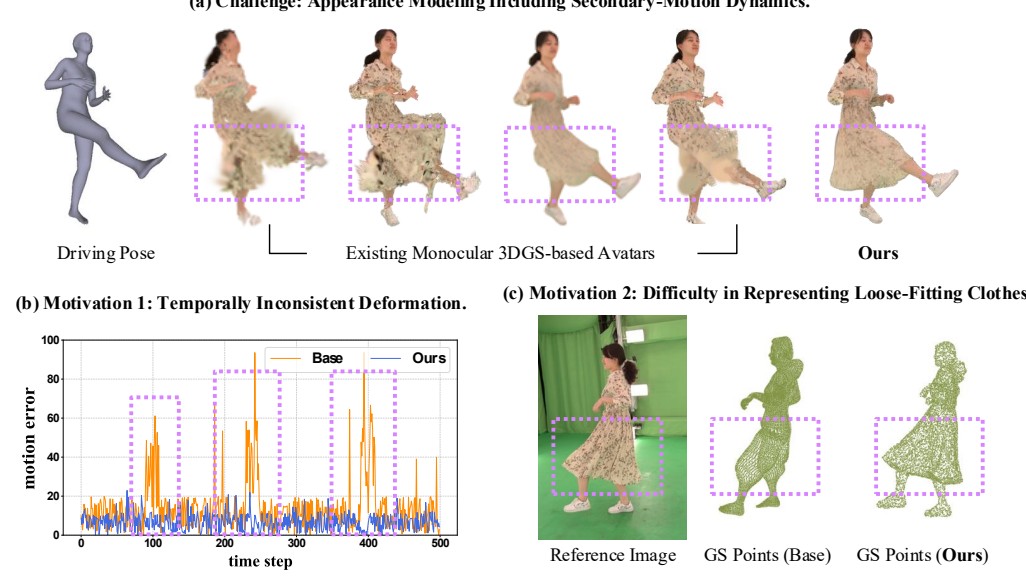

**Figure 1:** Conventional 3DGS-based avatars (Base) fail to model the dynamic appearance of subjects wearing loose garments, particularly in scenarios involving secondary motion (Lei et al., 2024; Qian et al., 2024b; Hu et al., 2024a; Moon et al., 2024). (a) Our method overcomes these limitations, enabling high-fidelity rendering of Gaussian avatars exhibiting dynamic motion from a single video. (b) In contrast, conventional methods define deformation through pre-specified articulation models such as linear blend skinning, which operate independently on each frame without accounting for temporal context—often leading to motion error spikes that indicate poor alignment with driving signals. (c) Furthermore, for initializing 3D Gaussians, they rely on a parametric template model resembling a naked body, which struggles to generate Gaussians for clothing regions deviating from the body surface, especially with loose-fitting garments.

Secondly, explicit representations based on 3D Gaussians are highly sensitive to the accuracy of their initial point placement, as widely discussed in primary 3DGS studies (Yu et al., 2024; Luiten et al., 2024). However, existing methods for creating 3DGS-based avatars from monocular videos (Lei et al., 2024; Hu et al., 2024a; Moon et al., 2024; Qian et al., 2024b), rely on parametric template models to initialize the shape of articulated subjects. These template models represent a naked body shape, leading to significant discrepancies between the initialized points and the actual shape when dealing with subjects wearing loose-fitting garments (Fig. 1c). Therefore, a few Gaussians should represent not only the body parts but also the appearance of the clothing, causing artifacts in novel pose animations where the model has not observed similar poses during training.

In this paper, we present a novel framework for modeling dynamic appearances of loose-fitting garments in 3D avatars, explicitly addressing the challenges posed by secondary motion. Our approach introduces a two-stage process, starting with personalized Gaussian initialization to accurately align Gaussian primitives with clothed silhouettes without relying on restrictive naked-body templates. Central to our system is the Secondary Motion-Aware Gaussian Deformation (SMAD) module, which constructs a velocity-encoded Gaussian graph and autoregressively predicts second-order Gaussian dynamics. By moving beyond the limitations of skeletal linear blend skinning, SMAD enables realistic modeling of fine-grained cloth motion—such as inertia-driven flutter—while strictly preserving structural coherence during deformation to ensure high-fidelity rendering across diverse and unseen poses.

Our contributions are summarized as follows:

- We propose a novel method for animatable 3D avatar reconstruction based on 3DGS, which enables dynamic appearance modeling of dressed avatars.

- We propose a secondary motion-aware Gaussian deformation, introducing a velocity-encoded Gaussian graph representation that autoregressively estimates Gaussian dynamics.

- Extensive experiments demonstrate that our method outperforms existing methods on the subjects wearing loose-fitting clothes with dynamic movement.

## 2 RELATED WORK

**Animatable 3D Avatars from Multi-view Videos.** It has long been a major focus in vision and graphics. Early systems (Stoll et al., 2010; Alldieck et al., 2018; Joo et al., 2015; Pons-Moll et al., 2017; Habermann et al., 2019) reconstructed actors in multi-view studios and animated meshes via multi-view geometry and hand-crafted articulation designs. While these approaches empower the controllability, it required substantial expert intervention. The shift to implicit neural representations, especially neural radiance fields (Mildenhall et al., 2021), introduced photorealistic neural avatars (Peng et al., 2021b;a; Habermann & et al., 2021; Zheng et al., 2023; Shen et al., 2023b; Li et al., 2023; Zhu et al., 2024; Shen et al., 2023a; Yin et al., 2023b; Chen et al., 2024; Saito et al., 2024) and free-view synthesis (Kwon et al., 2021; Liu et al., 2021; Işık et al., 2023; Kwon et al., 2024b), though often with slow training and additional structural constraints for stable driving. The 3DGS (Kerbl & et al., 2023) further achieved efficient rendering with high fidelity (Li et al., 2024; Zielonka et al., 2025; Zheng et al., 2024; Kwon et al., 2024a; Lin et al., 2024; Zhan et al., 2025; Liao et al., 2024). Yet, their high-fidelity performance fundamentally relies on dense, calibrated multi-view supervision, geometry constraints, and explicit subject-specific ground-truth template meshes. By contrast, our method is deliberately designed for the single-video setting, aiming to create user-friendly animatable clothed avatars directly from casual monocular footage.

**Animatable 3D Gaussian Avatars from Monocular Videos.** Advances in neural rendering and markerless motion-capture techniques have enabled the construction of user-friendly 3D avatars from monocular videos. With these advancements, it has been to learn a neural implicit representation defined in a continuous canonical space near a template mesh, and to deform this representation into the observation space using predefined articulations driven by motion inputs (Su et al., 2021; Weng et al., 2020; Chen et al., 2021; Weng et al., 2022; Wang et al., 2022; Yu et al., 2023; Jiang et al., 2023a;b). The advent of 3DGS has further accelerated photo-realistic modeling of 3d avatars; several works attach Gaussian primitives to a skeletal model and learn pose-conditioned deformations from monocular videos (Qian et al., 2024b; Hu et al., 2024a; Moon et al., 2024; Shao et al., 2024; Lei et al., 2024; Hu et al., 2024b; Zhai et al., 2025; Guo et al., 2025). However, existing methods assume template (Loper et al., 2015)-based initialization and its pre-defined articulation, which struggle to capture subtle, temporally coherent non-rigid effects. We build upon other line, introducing two key aspects: a template-free initialization that directly aligns the Gaussian primitives, eliminating the need for naked-body templates; and a physics-inspired autoregressive deformation module that predicts velocities and accelerations with finite difference method (Xie et al., 2024), and captures second-order dynamics through a velocity-encoded Gaussian graph.

**Dynamic Clothed Human Modeling.** Beyond primary motion driven by the main body movement, several works Ma et al. (2021; 2022); Prokudin et al. (2023); Seth et al. (2025) have also considered secondary motion, such as cloth dynamics. One line of works (Habermann et al., 2020; Habermann & et al., 2021; Habermann et al., 2021; Liao et al., 2024; Feng et al., 2022; 2023; Guo et al., 2023; 2024) reconstructs clothed surfaces using neural implicit representations, but canonicalization with predefined articulation struggles with loose garments. Another approaches combines non-rigid deformation with LBS and neural networks, yet requires subject-specific ground-truth clothed meshes. A separate direction incorporates physics simulation by numerically solving differential equations of the dynamic systems (Terzopoulos et al., 1987; Müller et al., 2007; Macklin et al., 2016). However, they are computationally expensive and difficult to parametrize; To alleviate this, several works (Pan et al., 2022; Santesteban et al., 2022; Grigorev et al., 2023; 2024) approximate dynamic systems with neural networks—e.g., augmenting models with virtual bones or employing recurrent architectures to predict garment-deformation sequences, but depend on high-level supervision such as 4D scans, cloth–body segmentation, and explicit colliders, and operate only on polygonal meshes. Our work is different lines of this work; we aim to create avatars represented as 3D Gaussian primitives and model its dynamics and appearances, given only from monocular videos, without access to any 3D ground-truth or prior geometric knowledge.

## 3 METHOD

Given a monocular RGB video $\mathcal{V} = \{I_t\}_{t=1}^{T}$ capturing a human subject in motion, our goal is to reconstruct a fully animatable 3D Gaussian avatar that faithfully models dynamic appearances of

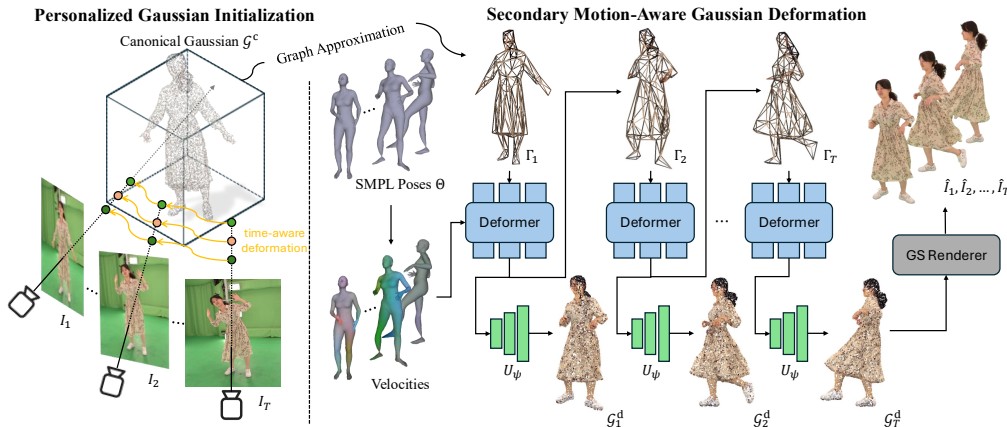

**Figure 2:** To model secondary motions in 3DGS-based avatars, we introduce a two-stage framework: (1) Personalized Gaussian Initialization using a deformable NeRF to estimate canonical Gaussians $\mathcal{G}^{\text{c}}$, and (2) Secondary Motion-Aware Deformation. $\mathcal{G}^{\text{c}}$ are structured as a Gaussian graph $\Gamma$, processed by a GNN-based autoregressive deformer, and decoded via $U_\psi$ into deformed Gaussians $\mathcal{G}^{\text{d}}$. Motion descriptors derived from SMPL poses $\Theta$ guide temporally coherent deformation. Then GS Renderer then synthesizes the final images.

loose-fitting clothed subjects. We adopt a dynamic set of 3D Gaussian primitives whose spatiotemporal properties evolve over time to capture complex non-rigid deformations, such as clothing dynamics. Formally, we represent the avatar at time $t$ as a set of $N$ deformed Gaussians:

$$\mathcal{G}_t^{\text{d}} = \{(\mu_{t,i}, \Sigma_{t,i}, c_{t,i}, \alpha_{t,i})\}_{i=1}^N,\tag{1}$$

where $\mu_{t,i} \in \mathbb{R}^3$ is the 3D mean position, $\Sigma_{t,i} \in \mathbb{R}^{3\times 3}$ is the covariance matrix modeling spatial extent and orientation, $c_{t,i} \in \mathbb{R}^3$ denotes RGB color, and $\alpha_{t,i} \in \mathbb{R}$ represents opacity. The number of primitives $N$ remains fixed across time, but their parameters are dynamically updated to reflect conditioning motion, a set of SMPL poses $\Theta = \{\theta_t\}_{t=1}^T$. We then obtain the animated rendering video $\hat{\mathcal{V}} = \{\hat{I}_t = \mathcal{R}(\mathcal{G}_t^{\text{d}})\}_{t=1}^T$ by projecting the deformed Gaussians through a differentiable splatting renderer $\mathcal{R}$. Fig. 2 illustrate the overall process of the proposed method.

**Baselines.**   We adopt a simple baseline that obtains dense canonical Gaussian primitives using a 4D NeRF (Pumarola et al., 2021). Concretely, we train a deformable neural radiance field on the input monocular video and map each observation-space point $\mathbf{x}_t$ at time $t$ to a canonical space (reference time). By querying color and density in the canonical space, we recover a dense canonical density field that captures both body and loose clothing without relying on a parametric template.

After training, we extract canonical Gaussians by thresholding the time-averaged canonical density $\bar{\sigma}(\mathbf{x}) = \frac{1}{T}\sum_t \sigma(\mathbf{x}, t)$ and clustering the surviving voxels to obtain Gaussian centers $\{\boldsymbol{\mu}_i^{\text{c}}\}$, with isotropic variances $\{\Sigma_i^{\text{c}}\}$ and colors $\{\mathbf{c}_i^{\text{c}}\}$. This yields a dense set of canonical 3D Gaussian primitives $\mathcal{G}^{\text{c}}$ that serves as the person-specific Gaussian initialization (PGI) for subsequent stages.

### 3.1 VELOCITY-ENCODED GAUSSIAN GRAPH

To overcome the limited capability of representing secondary motions caused by reliance on linear blend skinning of parametric template human models (Loper et al., 2015; Pavlakos et al., 2019), we propose an autoregressive Gaussian deformation method that moves beyond the template model. Furthermore, to ensure robust performance even when the number of Gaussians grows exponentially and to alleviate computational complexity, we propose a graph-based deformation approach that approximates Gaussian interactions.

**Graph Construction.** Given a set of $N$ initial Gaussian points $\{\mu_1, \mu_2, ..., \mu_N\}$, we downsample the Gaussian points to $\mathbf{X} \in \mathbb{R}^{M\times 3}$ ($M \ll N$) with voxel-grid downsampling (Rusu & Cousins, 2011); these $M$ nodes serve as the final Gaussian primitives used for rendering. We then construct a graph $\Gamma = (\mathbf{X}, \mathbf{A})$, where $\mathbf{A} \in \mathbb{R}^{M\times M}$ is adjacency matrix. It is constructed via $k$-Nearest Neighbors ($k$-NN) by computing pairwise distances $\text{d}(\mathbf{x}_i, \mathbf{x}_j) = \|\mathbf{x}_i - \mathbf{x}_j\|_2, \quad \forall \mathbf{x}_i, \mathbf{x}_j \in X$. Each element of

$\mathbf{A}_{ij}$ is formulated as $\exp\left(-\frac{\mathrm{d}(\mathbf{x}_i, \mathbf{x}_j)^2}{\rho_a^2}\right)$, where $\rho_a$ controls sensitivity to distances.

**Velocity Encoding (VE).** We build the node features $\mathbf{H} = \{\mathbf{h}_1, \mathbf{h}_2, ..., \mathbf{h}_M\}$ at each node position $\mathbf{x}_i$. Let us consider $\mathbf{h}_i$ as a concatenation of the node position $\mathbf{x}_i$ and its velocity $\mathbf{v}_i(t) = \frac{\mathbf{x}_i(t) - \mathbf{x}_i(t - \Delta t)}{\Delta t}$ at the time state $t$. Furthermore, to capture long-range dependencies, we buffer the past $\tau_v$ memory vectors as a set of $\bar{\mathbf{v}}_i = \{\mathbf{v}_i(t), \mathbf{v}_i(t - 1), \ldots, \mathbf{v}_i(t - \tau_v)\}$ To condition a set of body pose priors $\Theta_{t-\tau:t} = \{\theta_t, \theta_{t-1}, ..., \theta_{t-\tau}\}$ with time window $\tau$, we additionally embed it to $\mathbf{E} = \{\mathbf{e}_1, \mathbf{e}_2, ..., \mathbf{e}_M\}$ with MLP. At this end, the node feature $\mathbf{h}_i$ is defined as $\mathbf{h}_i = (\mathbf{x}_i, \bar{\mathbf{v}}_i, \mathbf{e}_i)$.

## 3.2 SECONDARY MOTION-AWARE DEFORMATION (SMAD)

Our goal is to move beyond linear blend skinning (LBS) with parametric body models (Loper et al., 2015) and learn an animatable 3DGS avatar that can faithfully reproduce secondary motions. Motivated by deformation methods that generalize to unseen motions without relying on pre-defined kinematic hierarchies (Zheng et al., 2021; Grigorev et al., 2023), we employ a graph neural network (GNN) deformer that autoregressively predicts the non-rigid dynamics of human bodies—hence, of Gaussian primitives.

**Definition.** We model each Gaussian node $i$ as a point mass $g_i$ whose motion follows a second-order mass–spring–damper system (Terzopoulos et al., 1987; Provot et al., 1995). Let $\mathbf{x}_i(t) \in \mathbb{R}^3$ and $\mathbf{v}_i(t) = \dot{\mathbf{x}}_i(t) \in \mathbb{R}^3$ denote the position and velocity at time $t$. The dynamics are

$$\mathbf{F}_i^{\text{ext}}(t) = g_i\,\ddot{\mathbf{x}}_i(t) + \gamma_i\,\dot{\mathbf{x}}_i(t) + \sum_j k_{ij}\left(\mathbf{x}_i(t) - \mathbf{x}_j(t) - \mathbf{L}_{ij}^{\text{rest}}\right), \tag{2}$$

where $\ddot{\mathbf{x}}_i(t) = \mathbf{a}_i(t)$ is acceleration, $\gamma_i$ is a damping coefficient, $k_{ij}$ is the spring stiffness between nodes $i$ and $j$, $\mathbf{L}_{ij}^{\text{rest}}$ is their rest offset in canonical space, and $\mathbf{F}_i^{\text{ext}}(t)$ is an external driving force.

With a discrete step $\Delta t$, we apply explicit Euler integration:

$$\mathbf{a}_i(t) = \frac{1}{g_i}\left(\mathbf{F}_i^{\text{ext}}(t) - \gamma_i\,\mathbf{v}_i(t) - \sum_j k_{ij}\left[\mathbf{x}_i(t) - \mathbf{x}_j(t) - \mathbf{L}_{ij}^{\text{rest}}\right]\right), \tag{3}$$

$$\mathbf{v}_i(t + \Delta t) = \mathbf{v}_i(t) + \Delta t\,\mathbf{a}_i(t), \qquad \mathbf{x}_i(t + \Delta t) = \mathbf{x}_i(t) + \Delta t\,\mathbf{v}_i(t + \Delta t). \tag{4}$$

This second-order formulation naturally induces secondary motion (e.g., cloth flutter). In practice, we let a message-passing GNN (Gilmer et al., 2020) learn these updates rather than prescribing forces explicitly.

**Architecture.** The Gaussian graph deformer parameterizes the above updates with a message-passing GNN. Each node $i$ carries a feature $\mathbf{h}_i(t) \in \mathbb{R}^{d_{\text{h}}}$ obtained from Sec. 3.1. At time $t$, node $i$ aggregates information from its neighbors using an adjacency $\mathbf{A}_{ij}(t)$. With an MLP $M_\theta$, we define messages as

$$\mathbf{m}_{j \to i}(t) = M_\theta\left(\mathbf{h}_i(t), \mathbf{h}_j(t)\right) \in \mathbb{R}^{d_{\text{m}}}, \qquad \mathbf{m}_i^{\text{agg}}(t) = \sum_j \mathbf{A}_{ij}(t)\,\mathbf{m}_{j \to i}(t). \tag{5}$$

Two update functions then produce the next-step node feature and physical state:

$$\mathbf{h}_i^\ell(t) = F_\theta\left(\mathbf{h}_i^{\ell-1}(t), \mathbf{m}_i^{\text{agg}}(t)\right), \tag{6}$$

$$\mathbf{a}_i(t) = G_\theta\left(\mathbf{h}_i^\ell(t), \mathbf{m}_i^{\text{agg}}(t)\right), \tag{7}$$

where $G_\theta$ serves as a neural surrogate for the mass–spring–damper updates in Eq. 2. After passing through final message-passing layers, we obtain updated positions and velocities for all nodes. Each node corresponds to a deformed Gaussian $\mathcal{G}_i^d$, and we finally set:

$$\boldsymbol{\mu}_i \leftarrow \mathbf{x}_i(t + \Delta t), \qquad \mathbf{c}_i,\ \alpha_i,\ \Sigma_i \leftarrow U_\psi\left(\mathbf{z}_i, \mathbf{h}_i^\ell(t)\right), \tag{8}$$

where $\mathbf{z}_i \in \mathbb{R}^{d_z}$ is a learned latent code and $U_\psi$ predicts color, opacity, and covariance for each Gaussian.

**Training Objectives.** After computing $\mathcal{G}^{\mathrm{d}} = \{\mathcal{G}_1^{\mathrm{d}}, \mathcal{G}_2^{\mathrm{d}}, ..., \mathcal{G}_T^{\mathrm{d}}\}$, we render it via Gaussian Splatting-based rasterizer $\mathcal{R}$ to $\hat{I}_t = \mathcal{R}(\mathcal{G}_t^{\mathrm{d}})$. We define a total SMAD loss term $\mathcal{L}_{\mathrm{SMAD}}$ as:

$$\mathcal{L}_{\mathrm{SMAD}} = \mathcal{L}_{\mathrm{RGB}} + \lambda_{\mathrm{iso}}\mathcal{L}_{\mathrm{iso}} + \lambda_{\mathrm{damp}}\mathcal{L}_{\mathrm{damp}}. \tag{9}$$

We mainly use the common L1 rgb photometric loss between rendered images and ground-truth images, which is formulated as:

$$\mathcal{L}_{\mathrm{RGB}} = \left\| \mathcal{R}(\mathcal{G}_t^d) - I_t \right\|_1. \tag{10}$$

It minimizes the pixel intensities of rendered Gaussians $\mathcal{R}(\mathcal{G}_t^d)$ to the ground-truth images $I_t$. In addition, we utilize two regularization terms:

$$\mathcal{L}_{\mathrm{iso}} = \sum_{(i,j)\in\mathcal{E}} \left( \|\mathbf{x}_i - \mathbf{x}_j\|_2 - \left\| \mathbf{L}_{ij}^{\mathrm{rest}} \right\|_2 \right)^2, \quad \mathcal{L}_{\mathrm{damp}} = \sum_{i=1}^{N}\sum_{t=1}^{T} \|\mathbf{v}_i(t)\|_2^2. \tag{11}$$

The isometry Loss $\mathcal{L}_{\mathrm{iso}}$ penalizes deviations in geodesic distance to preserve local surface area. It prevents stretching or shrinking of garment regions; useful for preserving cloth realism during motion. We set $\lambda_{\mathrm{iso}} = 0.1$, where it emphasize length preservation. The damping Loss $\mathcal{L}_{\mathrm{damp}}$ regularizes velocity magnitudes to reduce high-frequency vibration and dynamic instability. It reduces visual fluttering or noise in motion, especially noticeable in fine cloth edges. We $\lambda_{\mathrm{damp}} = 0.01$, where it avoids over-constraining dynamic details.

## 4 EXPERIMENTS

**Dataset.** **ZJU-MoCap** (Peng et al., 2021a) is a primary benchmark for animatable 3D avatars. Using HumanNeRF (Weng et al., 2022) split sequences, we report novel view synthesis results due to limited pose diversity. To supplement the lack of motion variation and loose-fitting garments, we additionally evaluate on two benchmarks. **4D-Dress** (Wang et al., 2024a) firstly introduces real-world 4D human clothing dataset featuring dynamic cloth motions, designed to advance research in realistic garment modeling and animation. We carefully selected five subjects, each wearing loose-fitting clothing such as skirts or puffer jackets. We also introduce **LoCo-Human**, a new in-the-wild dataset featuring five **Lo**ose-**Clo**thed **Human**s performing 5 dynamic and 1 static motions per subject. The static sequence is used for training, and the others for evaluating generalization in-the-wild scenarios.

**Baselines & Evaluation Metrics.** We compare our method with existing approaches on 3DGS-based avatars from monocular videos. Given the extensive body of prior work in this domain, it is practically infeasible to compare against all existing methods. Therefore, we specifically focus on publicly available baseline methods (Lei et al., 2024; Hu et al., 2024a; Qian et al., 2024b; Moon et al., 2024) that explicitly address dynamic appearance modeling. We evaluate the visual fidelity of the rendered animatable avatars with widely used metrics: PSNR, SSIM, and LPIPS. PSNR and SSIM measure pixel-level similarity and structural consistency with the ground-truth images, while LPIPS captures perceptual quality based on deep feature distances. These metrics collectively assess both low-level accuracy and high-level perceptual realism. To quantitatively assess both temporal consistency and how faithfully the animated avatars follow the driving motion, we compute the motion error (Kanazawa et al., 2019) between driving signal and motion estimated from the rendered animations.

### 4.1 RESULTS

We comprehensively compare our proposed method to state-of-the-art animatable 3D Gaussian avatar methods on three datasets: the 4D-Dress, ZJU-Mocap, and our introduced LoCo-Human in-the-wild evaluation dataset. We adopt standard metrics including PSNR, SSIM, and LPIPS to quantitatively measure visual fidelity and perceptual quality of animated avatars in rendered images. We first evaluate the ability to synthesize novel poses of dressed avatars on 4D-Dress. As presented in Tab. 1 (a), our method outperforms baseline methods across all metrics, demonstrating superior reconstruction quality. Qualitative comparisons (Fig. 3) further confirm our method's capability to produce realistic cloth dynamics, mitigating common artifacts such as unrealistic garment splitting observed in baselines. Next, we formally compare our method on the widely-used ZJU-Mocap benchmark. Following the conventional evaluation protocol, we quantitatively and qualitatively assess novel-view synthesis quality (Tab. 1 (b) & Fig. 5). Results indicate that our method consistently achieves superior

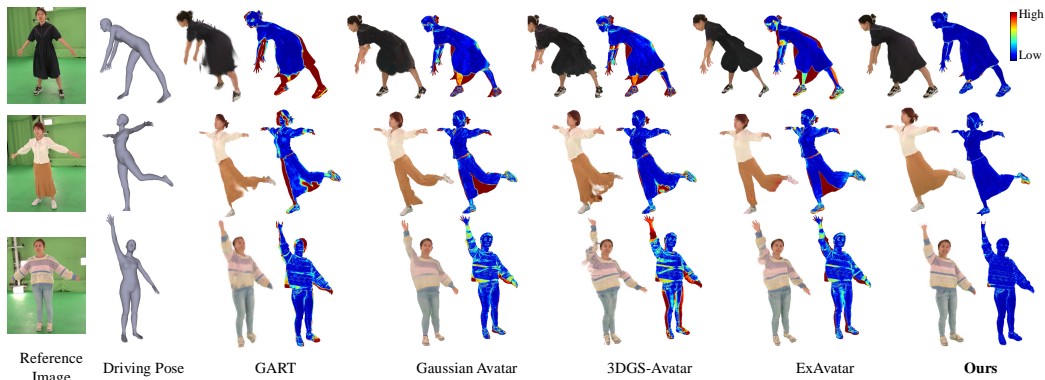

**Figure 3:** Qualitative comparison of novel pose synthesis on 4D-Dress dataset. We compare our method to the serveral 3D Gaussian Splatting-based Avatars (Lei et al., 2024; Hu et al., 2024a; Qian et al., 2024b; Moon et al., 2024). For each subject, we present reference image, driving pose, rendered image and error map to ground-truth image. Our method models robust dynamic appearances wearing loose-fitting clothes compared to the baselines.

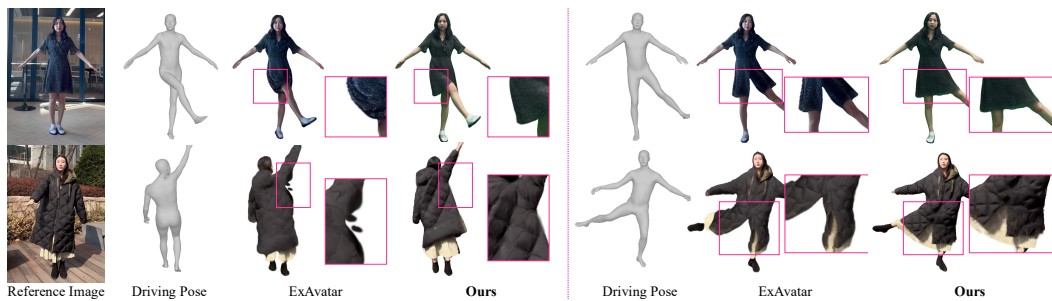

**Figure 4:** Qualitative comparison on the in-the-wild LoCo-Human dataset. Given the target driving poses, we animate avatars wearing loose-fitting garments. Compared to ExAvatar (Moon et al., 2024), our method better preserves cloth details and faithfully produces coherent motion under diverse poses. Insets highlight finer garment structures, showing sharper appearances and more realistic deformation.

performance compared to previous single-video avatar approaches, reflecting improvements in visual sharpness and perceptual realism. In addition, we conduct extended evaluations on LoCo-Human, an in-the-wild dataset to assess the generalization ability of our method in real-world scenarios. As shown in Tab. 1 (c), our approach consistently outperforms existing baselines across diverse subjects. The qualitative results Fig. 4, further support these findings—demonstrating the robustness of our method even in scenarios involving challenging clothing, complex motions, and various confounding factors. These empirical results suggest that our deformation network, which mimics a second-order dynamic system, better captures cloth dynamics compared to conventional deformation schemes based on parametric template models. These extensive experiments validate that our approach effectively addresses critical challenges associated with dynamic appearance modeling from single monocular videos.

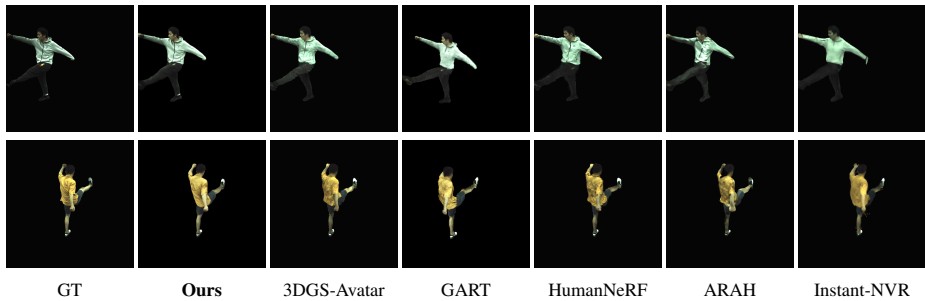

**Figure 5:** Qualitative comparison of novel view synthesis on ZJU-Mocap (Peng et al., 2021a). Our method yields view-consistent and artifact-free appearance modeling, even for repetitive motions in novel view synthesis scenarios.

**(a) Novel Pose Synthesis on 4D-Dress Dataset**

| Method | 00148 | | | 00170 | | | 00185 | | | 00187 | | | 00190 | | |
|---|---|---|---|---|---|---|---|---|---|---|---|---|---|---|---|
| | PSNR↑ | SSIM↑ | LPIPS↓ | PSNR↑ | SSIM↑ | LPIPS↓ | PSNR↑ | SSIM↑ | LPIPS↓ | PSNR↑ | SSIM↑ | LPIPS↓ | PSNR↑ | SSIM↑ | LPIPS↓ |
| GART (Lei et al., 2024) | 20.86 | 0.9509 | 0.0661 | 23.52 | 0.9622 | 0.0413 | 26.84 | 0.9599 | 0.0488 | 25.81 | 0.9401 | 0.0592 | 29.01 | 0.9627 | 0.0375 |
| Gaussian Avatar (Hu et al., 2024a) | 20.91 | 0.9512 | 0.0657 | 24.12 | 0.9630 | 0.0356 | 26.62 | 0.9586 | 0.0500 | 24.96 | 0.9317 | 0.0684 | 26.44 | 0.9591 | 0.0512 |
| 3DGS-Avatar (Qian et al., 2024b) | _22.79_ | _0.9560_ | _0.0471_ | 25.49 | 0.9636 | **0.0293** | 27.54 | 0.9595 | _0.0394_ | _25.99_ | 0.9398 | _0.0457_ | **29.49** | _0.9616_ | **0.0278** |
| ExAvatar (Moon et al., 2024) | 21.93 | 0.9536 | 0.0628 | _26.30_ | _0.9657_ | 0.0367 | _28.35_ | _0.9618_ | 0.0470 | 25.84 | _0.9403_ | 0.0620 | 26.12 | 0.9586 | 0.0569 |
| **Ours** | **24.74** | **0.9601** | **0.0397** | **27.62** | **0.9700** | _0.0301_ | **29.98** | **0.9673** | **0.0370** | **27.71** | **0.9548** | **0.0443** | _29.44_ | **0.9635** | _0.0347_ |

**(b) Novel View Synthesis on ZJU-MoCap**

| Method | 394 | | | 393 | | | 392 | | | 387 | | | 386 | | |
|---|---|---|---|---|---|---|---|---|---|---|---|---|---|---|---|
| | PSNR↑ | SSIM↑ | LPIPS↓ | PSNR↑ | SSIM↑ | LPIPS↓ | PSNR↑ | SSIM↑ | LPIPS↓ | PSNR↑ | SSIM↑ | LPIPS↓ | PSNR↑ | SSIM↑ | LPIPS↓ |
| NeuralBody (Peng et al., 2021a) | 29.10 | 0.9593 | 0.0545 | 28.61 | 0.9590 | 0.0590 | 30.10 | 0.9642 | 0.0532 | 27.00 | 0.9518 | 0.0594 | 30.54 | 0.9678 | 0.0464 |
| HumanNeRF (Weng et al., 2022) | 30.31 | 0.9642 | 0.0328 | 28.31 | 0.9603 | 0.0367 | 31.04 | 0.9705 | 0.0321 | 28.18 | 0.9632 | 0.0355 | 33.20 | 0.9752 | 0.0289 |
| MonoHuman (Yu et al., 2023) | 29.15 | 0.9595 | 0.0380 | 27.64 | 0.9566 | 0.0431 | 29.50 | 0.9635 | 0.0394 | 27.93 | 0.9601 | 0.0417 | 32.94 | 0.9695 | 0.0360 |
| ARAH (Wang et al., 2022) | 29.46 | 0.9632 | 0.0407 | 28.77 | _0.9645_ | 0.0423 | _32.02_ | _0.9742_ | 0.0352 | _28.49_ | **0.9656** | 0.0404 | 33.50 | 0.9781 | 0.0314 |
| GART (Lei et al., 2024) | 29.92 | 0.9651 | 0.0325 | 28.65 | 0.9620 | 0.0355 | 31.36 | 0.9736 | 0.0305 | 28.20 | 0.9644 | 0.0344 | 33.48 | **0.9850** | 0.0295 |
| 3DGS-Avatar (Qian et al., 2024b) | _30.54_ | _0.9661_ | _0.0312_ | _28.88_ | 0.9635 | _0.0352_ | 31.66 | 0.9730 | _0.0301_ | 28.33 | 0.9642 | _0.0342_ | 33.63 | 0.9773 | _0.0257_ |
| **Ours** | **30.89** | **0.9677** | **0.0311** | **29.48** | 0.9643 | **0.0341** | **32.33** | **0.9754** | **0.0294** | **28.86** | _0.9650_ | **0.0329** | **33.86** | _0.9784_ | **0.0252** |

**(c) LoCo-Human (In-the-Wild)**

| Method | S01 | | | S02 | | | S03 | | | S04 | | | S05 | | |
|---|---|---|---|---|---|---|---|---|---|---|---|---|---|---|---|
| | PSNR↑ | SSIM↑ | LPIPS↓ | PSNR↑ | SSIM↑ | LPIPS↓ | PSNR↑ | SSIM↑ | LPIPS↓ | PSNR↑ | SSIM↑ | LPIPS↓ | PSNR↑ | SSIM↑ | LPIPS↓ |
| 3DGS-Avatar (Qian et al., 2024b) | 23.15 | 0.9374 | 0.0567 | 24.21 | 0.9391 | 0.0579 | 23.74 | 0.9349 | 0.0594 | 22.87 | 0.9337 | 0.0618 | 22.59 | 0.9312 | 0.0632 |
| ExAvatar (Moon et al., 2024) | _24.82_ | _0.9478_ | _0.0489_ | _25.07_ | _0.9483_ | _0.0468_ | _24.43_ | _0.9465_ | _0.0527_ | _23.93_ | _0.9442_ | _0.0543_ | _23.68_ | _0.9426_ | _0.0571_ |
| **Ours** | **26.17** | **0.9576** | **0.0423** | **26.44** | **0.9589** | **0.0409** | **25.76** | **0.9554** | **0.0441** | **25.38** | **0.9531** | **0.0467** | **24.83** | **0.9517** | **0.0484** |

**Table 1:** Quantitative comparisons across (a) novel pose synthesis on 4D-Dress, (b) novel view synthesis on ZJU-MoCap, and (c) LoCo-Human in-the-wild. We highlight the best (**bold**) and second-best (underline) performance in each case.

## 4.2 ABLATION STUDY

*Loss & Architecture Design.* We start from the base configuration of vanilla GNN with pose-dependent deformation, without any physically plausible information. Adding physics-inspired finite-difference method with its regularization (A1) yields a clear gain of +0.84 PSNR, and $-10.3\%$ LPIPS. Introducing an adaptive spring stifness $k_{ij}$ coefficients (A2), which adaptively distinguishes the rigid and non-rigid parts of subjects in unsupervised setting, further improves the rendering quality under dynamic motions. The advanced message-passing strategy for GNN with embedding edge features (A3) bring another boost of +0.68 PSNR, and $-10.2\%$ LPIPS. The full configuration with latent codes for time-varying dynamic appearance finally achieves the best results, which is +2.68 PSNR and a $31.0\%$ LPIPS reduction over A0, underscoring the complementary roles of physics constraints and graph design.

*Velocity Encoding (VE).* Encoding an autoregressive window of past velocities markedly improves temporal fidelity. Performance rises monotonically from no VE (B0) to larger horizons, peaking at $\tau_v = 11$ (B4), with a net +5.83 PSNR and a $40.3\%$ LPIPS drop. Very short context ($\tau_v = 1$) yields limited gains, while overly long horizons saturate; $\tau_v = 11$ strikes the best balance between temporal context and feature efficiency.

*SMAD Capacity (M).* Increasing the number of Gaussian graph nodes improves accuracy up to a moderate resolution. Compared to the no-SMAD baseline (C0), capacity scaling to $M = 40$k (C4) delivers +3.60 PSNR, +0.017 SSIM, and a $32.2\%$ LPIPS reduction. Extremely small graphs ($< 10$k) under-represent non-rigid dynamics, while very large ones (100k) underperform C4, suggesting optimization and overfitting issues at excessive capacity.

Fig. 6 (*left*) shows that VE reduces motion spikes by $35.5\%$, with green frames showing stable rendering and red frames showing flickering. Fig. 6 (*right*) shows PGI improves detail beyond the body, while SMAD removes skirt artifacts present in template-only results.

## 4.3 ANALYSES

*Model Selection.* To validate the effectiveness of our proposed design for SMAD module, we additionally conducted a controlled comparison against a carefully designed MLP-based autoregressive deformer, following (Zheng et al., 2021), and vanilla GNN. This baselines use the same inputs (positions, encoded velocities) as our method, ensuring a fair comparison. Table 4 shows that the MLP deformer fits the training motion but degrades significantly on unseen motion, while the

| Loss & Arch. Design | | | | Velocity Encoding (VE) | | | | SMAD Capacity ($M$) | | | |
|---|---|---|---|---|---|---|---|---|---|---|---|
| Method | PSNR↑ | SSIM↑ | LPIPS↓ | Method | PSNR↑ | SSIM↑ | LPIPS↓ | Method | PSNR↑ | SSIM↑ | LPIPS↓ |
| A0: Base | 25.21 | 0.952 | 0.058 | B0: w/o VE | 22.06 | 0.930 | 0.067 | C0: w/o SMAD | 24.29 | 0.946 | 0.059 |
| A1: + phys. reg | 26.05 | 0.956 | 0.052 | B1: $\tau_v = 1$ | 23.41 | 0.932 | 0.060 | C1: $M = 5$k | 25.36 | 0.950 | 0.053 |
| A2: + adaptive $k_{ij}$ | 26.44 | 0.958 | 0.049 | B2: $\tau_v = 7$ | 24.95 | 0.944 | 0.053 | C2: $M = 10$k | 26.47 | 0.958 | 0.048 |
| A3: + message-passing | 27.12 | 0.961 | 0.044 | B3: $\tau_v = 15$ | 26.78 | 0.958 | 0.045 | C3: $M = 100$k | 27.02 | 0.962 | 0.045 |
| **A4: Full (Ours)** | **27.89** | **0.963** | **0.040** | **B4: $\tau_v = 11$ (Ours)** | **27.89** | **0.963** | **0.040** | **C4: $M = 40$k (Ours)** | **27.89** | **0.963** | **0.040** |

**Table 2:** Ablation study on the effectiveness of our mainly proposed components. Three column blocks report (Left) loss/arch. design, (Middle) velocity encoding, and (Right) SMAD capacity $M$.

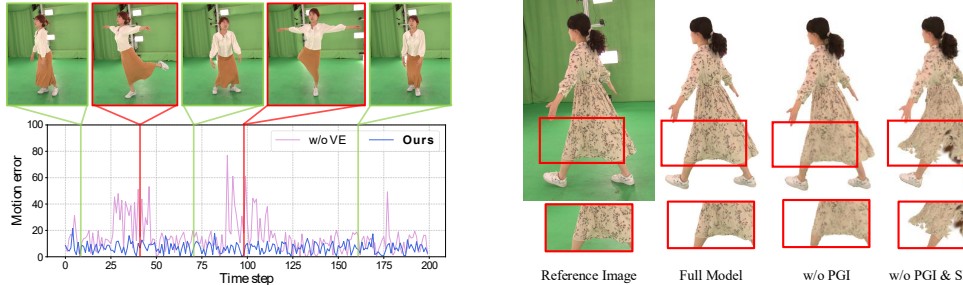

**Figure 6:** Ablation study on the visual effectiveness of (*left*) VE, (*right*) PGI, and SMAD. VE significantly reduces the motion error by encouraging temporal consistent deformation. PGI contributes to capturing fine-detailed clothing patterns, and SMAD sufficiently guarantees the robustness of clothing dynamics.

GNN-based deformers remains substantially more stable and accurate. This confirms that the graph-based formulation provides stronger structural priors and better generalization for clothed-human deformation. We also observe that embedding features on edges through message passing yields additional performance gains.

*Generalization.* Our auto-regressive deformation leverages a second-order state $(\mathbf{x}_t, \mathbf{v}_t)$, where velocities are obtained via finite differences. This provides a physically meaningful motion representation that aligns with how real deformable systems evolve, enabling more stable extrapolation than pose-only models. Prior work in human and cloth dynamics similarly shows that explicit velocity states improve rollout stability. By integrating over $(\mathbf{x}_t, \mathbf{v}_t)$ and regularizing with damping and local-isometry constraints, our model suppresses high-frequency drift and captures inertia-driven behavior, leading to robust generalization to unseen motions. Table 3 further supports this observation.

| Data | PSNR↑ | SSIM↑ | LPIPS↓ |
|---|---|---|---|
| *OOD* | 26.51 | 0.956 | 0.049 |
| *Test* | 27.89 | 0.963 | 0.040 |
| *Train* | 28.64 | 0.984 | 0.037 |

**Table 3:** Quantitative results on train/test, and out-of-distribution (OOD) motion sequences to evaluate generalization capability of our method (*blue*: p-val $p > 0.05$).

It reports quantitative results on the 4D-Dress subjects across train, test, and out-of-distribution (OOD) motion sequences. To assess whether performance differences across these distributions are statistically significant, we conducted paired t-tests for each setting. No comparison yielded a significant difference, indicating that our model maintains consistent performance regardless of motion distribution. This empirical evidence reinforces that our approach generalizes reliably to dynamic motions unseen during training.

*Error Accumulation.* It is well-known that auto-regressive models are prone to numerical error accumulation over long sequences. To analyze and reflect on this point, we captured two types of motion sequences, each lasting over 30 seconds: (a) a dynamic pose sequence, and (b) a repetitive pose sequence. We evaluated our proposed method, and also conducted a comparative analysis with and without our proposed velocity encoding scheme to evaluate its effectiveness. Our velocity encoding scheme appears to mitigate this issue by incorporating a history of multiple past states, rather than relying solely on the most recent estimate. This allows the model to remain robust even when the immediate past prediction is noisy, reducing the risk of cumulative drift.

*Training Cost.* Our model requires an average of 12.5 hours for personalized Gaussian initialization and 4.5 hours for training the secondary motion-aware deformation module, totaling approximately 17 hours on a single NVIDIA RTX 3090 GPU. Considering that existing state-of-the-art methods (Moon et al., 2024) typically require around 4 hours of training, our approach indeed incurs higher computa-

| Model | PSNR↑ | SSIM↑ | LPIPS↓ | PSNR↑ | SSIM↑ | LPIPS↓ |
|---|---|---|---|---|---|---|
| | | Test | | | Train | |
| MLP | 25.46 | 0.954 | 0.056 | 27.97 | 0.973 | 0.044 |
| vanilla GNN | 28.68 | 0.958 | 0.045 | 28.44 | 0.980 | 0.040 |
| **Ours** | 27.89 | 0.963 | 0.040 | 28.64 | 0.984 | 0.037 |

**Table 4:** Quantitative results of difference design choices of SMAD on train/test distributions.

| Model | PSNR↑ | SSIM↑ | LPIPS↓ | PSNR↑ | SSIM↑ | LPIPS↓ |
|---|---|---|---|---|---|---|
| | | (a) dynamic pose | | | (b) repetitive pose | |
| w/o VE | 24.47 | 0.949 | 0.050 | 24.69 | 0.950 | 0.049 |
| w/ VE (**Ours**) | **25.65** | **0.955** | **0.044** | **26.84** | **0.960** | **0.039** |

**Table 5:** Analyses of error accumulation ablating the velocity encoding (VE) strategy on two long sequences.

tional cost. However, we emphasize that, unlike prior methods whose limited model capacity yields only marginal gains even with extended training, our formulation continues to deliver significant performance improvements when trained longer (see Fig. 7). This suggests that our method possesses a higher effective capacity and is well suited for high-fidelity dynamic appearance modeling in personalized avatar reconstruction.

## 5 DISCUSSION

*On the Importance of Gaussian Initialization.* Accurate initialization is fundamental for animatable 3D Gaussian avatars, especially when modeling loose-fitting clothing. Prior monocular methods rely on parametric template bodies (Loper et al., 2015; Pavlakos et al., 2019), assuming minimally clothed geometry. As seen in Fig. 1(c), this creates large mismatches between template surfaces and real garment volumes, causing undersampling, silhouette distortion, and instability under unseen poses. Because Gaussians are explicit point samples, such errors propagate into deformation and cannot be repaired by skinning alone. Our personalized Gaussian initialization avoids these issues by estimating a clothed canonical field via neural deformable field, producing a subject-specific and geometry-faithful Gaussian distribution. This reduces the deformation network's burden, enabling it to focus on true non-rigid motion rather than fixing incorrect geometry. Fig. 6 show that PGI improves clothing detail, reduces skirt-splitting, and stabilizes secondary motion. Overall, initialization is not a preprocessing step but a critical determinant of garment fidelity and temporal stability.

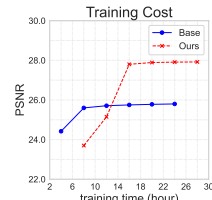

**Figure 7:** Analysis of the training-cost trade-off compared to an existing method.

*Template-free Deformation.* Even with a faithful canonical geometry, deformation remains constrained when tied to template-based articulation such as LBS. These methods define motion as a direct function of skeletal pose, which fails for loose garments that do not follow body kinematics. As visualized in Fig. 1(b), template-driven deformation creates motion-error spikes, flickering, and cloth splitting because it lacks temporal awareness and restricts non-rigid behavior. Our SMAD module departs from this paradigm by learning a template-free, autoregressive deformation field on a velocity-encoded Gaussian graph. Instead of following a fixed hierarchy, Gaussians interact through learned graph messages, enabling the model to infer how cloth regions co-move or lag independently of the body. This grants expressive, pose-agnostic deformation capability and yields coherent dynamics across diverse motions. Results in Table 2 and Fig. 6 show that removing template constraints dramatically improves robustness, generalization, and overall clothing realism.

## 6 CONCLUSION

In this paper, we introduced a novel approach for dynamic appearance modeling of 3D Gaussian Splatting-based avatars from monocular videos, focusing on loose-fitting clothing dynamics. We addressed two main challenges: limited Gaussian deformation from template articulation, and misalignment issues from Gaussian initialization relying on naked body templates. To resolve these, we proposed an autoregressive Gaussian deformation strategy that predicts velocities for realistic cloth dynamics, and a personalized Gaussian initialization using a deformable neural radiance field to capture clothed silhouettes. Additionally, we evaluate on an in-the-wild dataset featuring subjects performing dynamic movements in challenging clothing. Extensive evaluations confirmed our method improves realism and outperforms existing approaches in both controlled and unconstrained settings.

ACKNOWLEDGEMENTS

This research was supported by the Korea Creative Content Agency (KOCCA) grant funded by the Ministry of Culture, Sports and Tourism of the Republic of Korea (RS-2024-00441174), and supported by Institute of Information & communications Technology Planning & Evaluation (IITP) grant funded by the Korea government (MSIT) (RS-2019-II190079, Artificial Intelligence Graduate School Program (Korea University), and RS-2024-00457882, AI Research Hub Project).

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

# Appendix

## Table of Contents

# A NOTATION

We summarize the key notations used in main paper in Table F.

**Table F:** Summary of mathematical key notations.

| # | Notation | Dimension / Set | Definition |
|---|----------|-----------------|------------|
| 1 | $V = \{I_t\}_{t=1}^T$ | $I_t \in \mathbb{R}^{H \times W \times 3}$ | Monocular RGB video with $T$ frames |
| 2 | $T$ | $\mathbb{N}$ | Number of video frames |
| 3 | $\mathcal{G}_t^d = \{(\mu_{t,i}, \Sigma_{t,i}, c_{t,i}, \alpha_{t,i})\}_{i=1}^N$ | - | Deformed Gaussians at time $t$ |
| 4 | $N$ | $\mathbb{N}$ | Number of Gaussian primitives |
| 5 | $\mu_{t,i}$ | $\mathbb{R}^3$ | Position of the $i$-th Gaussian |
| 6 | $\Sigma_{t,i}$ | $\mathbb{R}^{3 \times 3}$ | Covariance matrix (size & orientation) |
| 7 | $c_{t,i}$ | $\mathbb{R}^3$ | RGB color |
| 8 | $\alpha_{t,i}$ | $\mathbb{R}$ | Opacity |
| 9 | $\Theta = \{\theta_t\}_{t=1}^T$ | $SO(3)^K$ | SMPL joint pose sequence |
| 10 | $\mathcal{R}(\cdot)$ | $(\mathbb{R}^3, \Sigma, \alpha)^N \to \mathbb{R}^{H \times W \times 3}$ | Differentiable 3D Gaussian renderer |
| 11 | $\mathcal{G}^c$ | - | Canonical (undeformed) Gaussian set |
| 12 | $\Gamma = (X, A)$ | $X \in \mathbb{R}^{M \times 3}, A \in \mathbb{R}^{M \times M}$ | Gaussian graph (nodes & adjacency) |
| 13 | $M$ | $\mathbb{N}$ | Number of downsampled graph nodes ($M \ll N$) |
| 14 | $x_i$ | $\mathbb{R}^3$ | Node positions after voxel downsampling |
| 15 | $A_{ij} = \exp\left[-\frac{\|x_i - x_j\|^2}{\rho_a^2}\right]$ | $[0, 1]$ | Edge weight (k-NN Gaussian kernel) |
| 16 | $\rho_a$ | $\mathbb{R}^+$ | Sensitivity factor for Gaussian kernel distance |
| 17 | $h_i = (x_i, \bar{v}_i, e_i)$ | $\mathbb{R}^{d_h}$ | Node feature: position, velocity buffer, pose embed |
| 18 | $e_i$ | $\mathbb{R}^{d_e}$ | Body pose prior embedding |
| 19 | $v_i(t) = \frac{x_i(t) - x_i(t - \Delta t)}{\Delta t}$ | $\mathbb{R}^3$ | Instantaneous velocity |
| 20 | $\tau_v$ | $\mathbb{N}$ | Past-velocity buffer length |
| 21 | $g_i$ | $\mathbb{R}^+$ | Point mass for physics model |
| 22 | $a_i(t) = \ddot{x}_i(t)$ | $\mathbb{R}^3$ | Acceleration of node $i$ |
| 23 | $\gamma_i$ | $\mathbb{R}^+$ | Damping coefficient |
| 24 | $k_{ij}$ | $\mathbb{R}^+$ | Spring stiffness between nodes $i, j$ |
| 25 | $L_{ij}^{\text{rest}}$ | $\mathbb{R}^3$ | Rest offset of the spring |
| 26 | $F_i^{\text{ext}}(t)$ | $\mathbb{R}^3$ | External force applied to node $i$ |
| 27 | $\Delta t$ | $\mathbb{R}^+$ | Simulation time-step |
| 28 | $m_{j \to i}(t)$ | $\mathbb{R}^{d_m}$ | Message passed from node $j$ to node $i$ |
| 29 | $z_i$ | $\mathbb{R}^{d_z}$ | Learnable latent code for Gaussian $i$ |
| 30 | $\bar{\sigma}(x) = \frac{1}{T} \sum_{t=1}^T \sigma(x, t)$ | $\mathbb{R}$ | Time-averaged density for Gaussian extraction |
| 31 | $\mathcal{L}_{iso}, \mathcal{L}_{damp}$ | - | Regularization loss terms |
| 32 | $\lambda_{iso}, \lambda_{damp}$ | $\mathbb{R}^+$ | Hyperparameters for loss weighting |

# B  IMPLEMENTATION DETAILS

## B.1  VELOCITY-ENCODED GAUSSIAN GRAPH

*Voxel-grid downsampling ($N \to M$).* Given a set of $N$ initial Gaussian points, we introduce autoregressively graph-based Gaussian deformation to transform the Gaussians without pre-defined articulation to template parametric model for enhancing secondary motion dynamics. To avoid an $\mathcal{O}(N^2)$ neighbourhood search and to limit graph size for the GNN, we down-sample the Gaussian cloud on an isotropic voxel grid:

1. *Grid resolution.* Let $d_{\min}$ denote the minimum distance below which two Gaussians would overlap in the 3DGS renderer (*e.g.* the renderer's splat radius at canonical scale). We choose the voxel edge length as $s = 2d_{\min}$, which empirically yields $\approx$10 Gaussians per occupied voxel.
2. *Hash insertion.* Every Gaussian is hashed into a voxel key. It retains the index with the *smallest* per-voxel rendering error, measured on a $4\times$ subsampled depth map; all other Gaussians in that voxel are discarded.
3. *Representative pooling.* For the surviving indices we conduct average pooling, giving a single *graph node*. The total number of nodes is $M = 40k$, an order of magnitude smaller than $N$ without noticeable quality loss.

*k-Nearest-Neighbour edge set.* With the down-sampled node coordinates $X = \{\mathbf{x}_j\}_{j=1}^M$ (which is different from the one defined in PGI), we build an undirected, symmetric $k$-NN graph: $A = \big\{ (i,j) \mid \mathbf{x}_i \in \mathrm{Top}k\big( \|\mathbf{x}_i - \mathbf{x}_j\|_2 \big) \big\}$. We set $k = 16$, which is sufficiently dense to preserve local manifold connectivity yet keeps the message-passing cost low.

## B.2  SECONDARY MOTION-AWARE GAUSSIAN DEFORMATION

**Architecture.** Given the velocity-encoded Gaussian graph $\Gamma = (X, A)$, SMAD converts the current node state at animation step $t$ into frame-specific Gaussian attribute deltas $\{\Delta\boldsymbol{\mu}_i, \Delta\boldsymbol{\Sigma}_i, \Delta\mathbf{c}_i, \Delta\alpha_i\}_{i=1}^M$, through three conceptually simple stages. We present the architectural details of our SMAD.

*Node projection.* Each node $i$ consists of concatenated features $\mathbf{h}_i = \{\mathbf{x}_i, \bar{\mathbf{v}}_i, \mathbf{e}_i\}$, where embedding vector through MLP is obtained by driving pose sequences $\Theta_{t-\tau:t} = \{\theta_t, ..., \theta_{t-\tau}\}$. The $\mathbf{h}_i$ seeds the message-passing stage.

*Message-passing iterations.* At each iteration $\ell$, we construct an edge feature vector for every directed pair $(i, j) \in A$

$$\mathbf{e}_{ij}^\ell = \big[ \mathbf{h}_i^{\ell-1},\ \mathbf{h}_j^{\ell-1},\ \bar{\mathbf{x}}_j - \bar{\mathbf{x}}_i,\ \bar{\mathbf{v}}_j - \bar{\mathbf{v}}_i,\ \|\bar{\mathbf{x}}_j - \bar{\mathbf{x}}_i\|_2,\ 1 \big].$$

The shared *edge-MLP* $M_\theta$ compresses $\mathbf{e}_{ij}^\ell$ to a high-dimensional message $m_{j \to i}(t)$. For every receiver node we perform mean aggregation over its $k$ nearest neighbours: $\mathbf{m}_i^{\mathrm{agg}} = \frac{1}{k} \sum_{j \in \mathcal{N}(i)} \mathbf{m}_{ij}$. The aggregated vector is fed, together with the previous hidden state $\mathbf{h}_i^{\ell-1}$, into a GRU cell $\mathbf{h}_i^\ell = G_\theta(\mathbf{m}_i^{\mathrm{agg}}, \mathbf{h}_i^{\ell-1})$. Because $F_\theta$ share weights across iterations, the network learns a recurrent, physics-inspired propagation of inertia without increasing parameter count.

*Decoder $U_\psi$.* After three iterations, we obtain the refined latent representation $h_i^\ell$ for each node. A two-stage MLP acts as a shared decoder whose final activations feed four independent linear heads:

$$\Delta\boldsymbol{\mu}_i,\ \Delta\mathbf{v}_i,\ \Delta\boldsymbol{\Sigma}_i,\ \Delta\mathbf{c}_i = U_\psi(h_i^\ell).$$

Here, $\Delta\boldsymbol{\mu}_i$ is a 3-D position offset, $\Delta\mathbf{v}_i$ a 3-D velocity refinement that is re-queued into the velocity ring buffer, $\Delta\boldsymbol{\Sigma}_i$ a log-diagonal covariance update, and $\Delta\mathbf{c}_i \in [0, 1]^3$ a colour residual (sigmoid-bounded). These deltas are added to the canonical attributes before the Gaussian splatting renderer is invoked for the current frame.

SMAD therefore (i) embeds pose and recent motion into a compact latent space, (ii) injects neighbourhood cues through three message-passing steps that emulate mass–spring–damper interactions, and (iii) decodes temporally coherent, view-aware adjustments to every Gaussian's geometry and appearance.

### B.3 TRAINING

We adopted 2-stage training. Fristly, we train personalized Gaussian initialization as pre-stage, where it locates initial Gaussians densely aligned onto the person-specific silhouette. Thereafter, we train secondary motion-aware Gaussian deformation to auto-regressively transform the canonical 3D Gaussians that are aware of clothing dynamics. We used the Adam optimizer (Kingma & Ba, 2015) with an initial learning rate of 0.001, decaying by a factor of 0.5 if no improvement is made in four consecutive epochs. We used PyTorch (Paszke et al., 2019) for the backend processing. All experiments were conducted on AMD Ryzen Threadripper PRO 5965WX CPU and an NVIDIA GeForce RTX 3090 GPU.

## C EXPERIMENTAL SETTING DETAILS

### C.1 DATASET DESCRIPTION

*Motivation.* Existing datasets for evaluating animatable 3D avatars predominantly focus on subjects wearing tight-fitting clothing and performing repetitive, often monotonous motions. Although the recently proposed 4D-Dress dataset (Wang et al., 2024b) addresses some of these limitations by including diverse garment types, it is still collected in a controlled laboratory setting and primarily designed for multi-view capture evaluations. However, our ultimate goal is to democratize avatar generation—making it robust and accessible to everyday users using only monocular inputs. To this end, it is essential to evaluate performance under in-the-wild scenarios, where diverse factors such as occlusion, motion blur, and uncontrolled lighting can affect avatar quality. We introduce an evaluation dataset **LoCo-Human**, featuring (1) subjects wearing loose-fitting garments, (2) realistic clothing dynamics exhibiting secondary motion, and (3) videos captured in the wild. This setting enables evaluation of avatar reconstruction robustness under real-world conditions.

*Dataset Statistics.* Our dataset comprises five unique subjects, each recorded in a total of six sequences: one static-motion sequence and five dynamic-motion sequences. For each subject, one sequence captures a 360-degree rotation, while the remaining four sequences feature free-form dynamic motion, performed without scripted guidelines. All subjects wear challenging garments, such as long skirts and padded coats, designed to emphasize loose-fitting clothing dynamics. Fig. H shows qualitative results on several samples from our dataset.

*Capture Setup.* All sequences were captured using standard smartphone devices. We provide 3 to 10-minute-long RGB video sequences for each subject, along with corresponding segmentation masks, depth maps, and SMPL pose parameters. Segmentation masks were obtained using SAM v2 (Ravi et al., 2024), with optional manual refinement to ensure silhouette accuracy. For depth information, we used smartphone depth cameras to obtain coarse estimates, which are further refined using Metric3D (Yin et al., 2023a). To extract driving pose parameters, we fit the SMPL model (Loper et al., 2015) to each input monocular video, following the protocol described in (Moon et al., 2024).

### C.2 BASELINE METHODS

Our objective is to construct animatable 3D human avatars from a single video, faithfully reflecting secondary motion, based on the 3D Gaussian Splatting representation. To validate the effectiveness of our method, we conduct a comparative analysis with existing 3DGS-based avatar approaches. Given the rapid expansion of research in this domain, an exhaustive comparison with all prior works is impractical. Therefore, we select a subset of publicly available methods, with particular emphasis on those that explicitly address dynamic appearance modeling.

*GART (Lei et al., 2024).* GART introduces the Gaussian Articulated Template (GART) model, designed to reconstruct non-rigid articulated human subjects from monocular videos. To facilitate challenging deformation, it employs a learnable forward skinning strategy via latent bones. However, due to the lack of supervision on where to place novel bones and how to assign skinning weights for each Gaussian, the method struggles to maintain structural consistency during deformation. This often leads to a breakdown in kinematic coherence and introduces excessive degrees of freedom, making stable animation difficult.

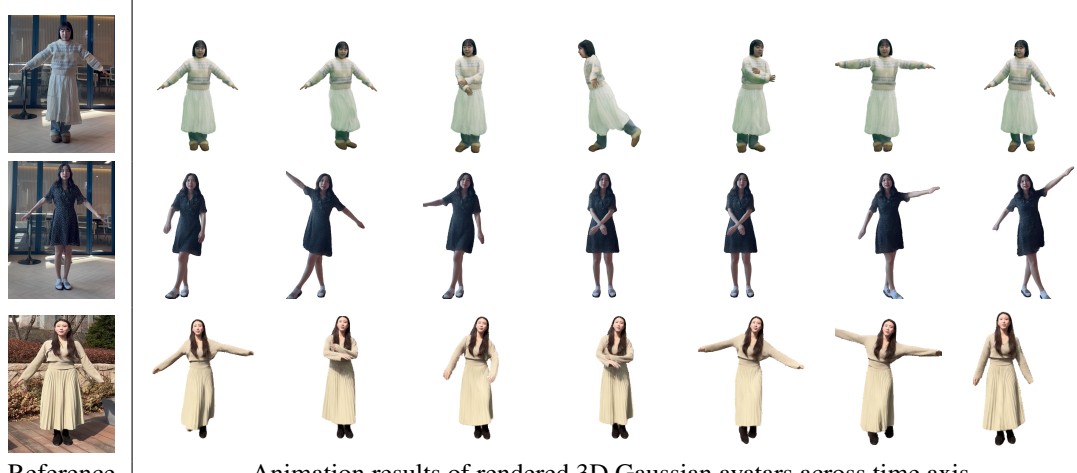

Reference | Animation results of rendered 3D Gaussian avatars across time axis.

**Figure H:** Qualitative results on LoCo-Human consisting of the subjects wearing loose-fitting clothes with dynamic motions.

*GaussianAvatar (Hu et al., 2024a).* GaussianAvatar proposes an efficient method for creating realistic human avatars with dynamic 3D appearances from a single video. It utilizes UV positional maps to encode pose-dependent features and integrates them with canonical surface geometry. However, its pose representation is heavily reliant on parametric template priors. While effective for minimally clothed humans, this reliance limits its generalization to clothed avatars whose geometry deviates significantly from the template, especially in the case of skirts, which often exhibit unnatural splitting between the legs under dynamic motion.

*3DGS-Avatar (Qian et al., 2024a).* 3DGS-Avatar also presents a framework for creating animatable human avatars from monocular video using 3D Gaussian primitives. It introduces a non-rigid deformation network that learns per-Gaussian offsets to represent dynamic clothed avatars. However, by assigning independent degrees of freedom to each Gaussian, the method neglects the underlying structural coherence of the avatar. This leads to undesired needle-like artifacts, particularly under dynamic motions.

*ExAvatar (Moon et al., 2024).* ExAvatar proposes a hybrid representation that combines a whole-body parametric mesh with 3D Gaussian Splatting. By binding each Gaussian to corresponding mesh vertices, the model ensures stable deformation under novel motions. Nevertheless, it exhibits weak appearance modeling for clothed subjects wearing loose-fitting garments, such as coats or skirts, which significantly deviate from the shape of the minimally clothed parametric template.

In summary, existing methods largely depend on shape and articulation priors from parametric template models to synthesize and animate avatars. This reliance limits their ability to model (1) loose-fitting clothed humans with geometry far from minimally clothed templates, and (2) realistic deformation that preserves geometric structure while capturing clothing dynamics. Our approach aims to overcome these limitations by introducing a template-free formulation tailored for secondary motion-aware avatar modeling.

## C.3 EVALUATION METRICS

We used PSNR, SSIM, LPIPS, and motion error as the primary evaluation metrics.

*Peak Signal-to-Noise Ratio (PSNR).* PSNR is a widely used metric for evaluating the reconstruction quality of compressed or reconstructed images by comparing them to the original. It quantifies the ratio between the maximum possible pixel value and the power of the distortion (error) introduced. Given an original image $I$ and a rendered image $\hat{I}$ of animatable 3D Gaussian avatars, we first compute the Mean Squared Error (MSE). Then the PSNR is defined as:

$$\text{PSNR} = 10 \cdot \log_{10}\left(\frac{255^2}{\text{MSE}}\right).$$

*Structural Similarity Index Measure (SSIM).* SSIM is a perceptual metric that quantifies image quality degradation based on changes in structural information, taking into account human visual perception. Unlike PSNR, it considers luminance, contrast, and structural similarity. Given two local image patches $x$ and $y$, SSIM is defined as:

$$\text{SSIM}(x, y) = \frac{(2\mu_x\mu_y + C_1)(2\sigma_{xy} + C_2)}{(\mu_x^2 + \mu_y^2 + C_1)(\sigma_x^2 + \sigma_y^2 + C_2)}$$

where $(\mu_x, \mu_y)$ are mean intensities, $(\sigma_x^2, \sigma_y^2)$ are variances, $\sigma_{xy}$ are covariance between $x$ and $y$. The final SSIM value is obtained by averaging local SSIM scores across the entire image.

*Learned Perceptual Image Patch Similarity (LPIPS).* LPIPS is a perceptual metric that compares images using deep features extracted from pretrained neural networks. It is designed to align closely with human perceptual judgments by evaluating similarity in a learned feature space. Given two images $I$ and $\hat{I}$, let $\hat{f}^l(x)$ and $\hat{f}^l(y)$ denote the normalized feature maps from layer $l$ of a pretrained network $\phi$, with spatial dimensions $H_l \times W_l$ and channel dimension $C_l$. Then LPIPS is defined as:

$$\text{LPIPS}(I, \hat{I}) = \sum_l w_l \cdot \frac{1}{H_l W_l} \sum_{h=1}^{H_l} \sum_{w=1}^{W_l} \left\| \hat{f}^l_{h,w}(I) - \hat{f}^l_{h,w}(\hat{I}) \right\|_2^2$$

where $w_l$ are learned weights that reweight the contribution of each layer to better match human perceptual similarity. We use deep features from (Simonyan & Zisserman, 2014). For ZJU-Mocap, following the convention of previous studies (Qian et al., 2024a), we reported the LPIPS values scaled by $10^3$ in the main draft to make the performance differences with the baselines more clearly distinguishable.

*Motion Error.* We additionally measured motion error to evaluate the temporally consistent animation and fidelity to the driving motion of the generated avatars. Specifically, this is computed as the acceleration error between the driving pose (used as the condition) and the corresponding pose of the rendered Gaussian avatar, which is estimated in reverse using a pre-trained pose estimator (Li et al., 2022). To assess this, we measured acceleration error, presented in (Kanazawa et al., 2019) the acceleration error metric is used. It measures the average deviation between the estimated and ground-truth joint accelerations across a temporal sequence. Given a sequence of 3D joint positions $\{\mathbf{x}^t_{\text{joints}} \in \mathbb{R}^{3J}\}_{t=1}^T$, the acceleration at time $t$ is approximated using the second-order finite difference: $\mathbf{a}^t_{\text{joints}} = \mathbf{x}^{t+1}_{\text{joints}} - 2\mathbf{x}^t_{\text{joints}} + \mathbf{x}^{t-1}_{\text{joints}}$. The acceleration error is then computed as:

$$\text{Motion Error} = \frac{1}{T-2} \sum_{t=2}^{T-1} \left\| \hat{\mathbf{a}}^t_{\text{joints}} - \mathbf{a}^t_{\text{joints}} \right\|_2$$

where $\hat{\mathbf{a}}^t_{\text{joints}}$ and $\mathbf{a}^t_{\text{joints}}$ denote the predicted and ground-truth joint accelerations, respectively.

# D  ADDITIONAL ANALYSES AND DISCUSSIONS

## D.1  HYPER-PARAMETER SEARCH

We ablated key hyperparameters: scaling factors of loss functions $\lambda_{\text{damp}} \in \{0.001\text{--}1.0\}$, and $\lambda_{\text{iso}} \in \{0.01\text{--}1.0\}$ in Fig. I. The results show that $\lambda_{\text{damp}} = 0.01$ and $\lambda_{\text{iso}} = 0.1$ performed best; $\lambda_{\text{damp}} = 1.0$ sharply degraded results, implying excessive temporal smoothing, while isotropy gains largely saturated past 0.1. Based on the results shown in the figure, both $\mathcal{L}_{\text{iso}}$ and $\mathcal{L}_{\text{damp}}$ help suppress excessive deformation and promote stable optimization, though they operate differently. $\mathcal{L}_{\text{iso}}$ preserves local isotropy, preventing geometric distortions, and its increase leads to a gradual improvement in PSNR. In contrast, $\mathcal{L}_{\text{damp}}$ mitigates excessive dynamic oscillations, yielding a more pronounced PSNR gain within an appropriate

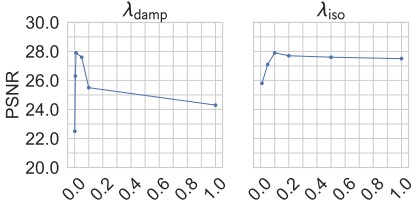

**Figure I:** Hyper-parameter search on weighting factor of loss functions $\lambda_{\text{damp}}$ and $\lambda_{\text{iso}}$.

range. Combined, the two terms jointly enforce structural fidelity and dynamic stability, achieving a balanced improvement in both visual consistency and numerical robustness.

## D.2 STATISTICAL SIGNIFICANCE

We conducted a two-sided paired $t$-test, conservatively setting $p = 0.05$ to relieve a multi-comparison issue. The test was based on SSIM metric scores evaluated across the test sequences of all subjects used for evaluation on the 4D-Dress. We performed (1) statistical significance analysis against comparison methods (Lei et al., 2024; Hu et al., 2024a; Qian et al., 2024b; Moon et al., 2024), and (2) significance testing against ablated versions of our proposed main components. When compared with 3DGS-based baseline methods, all resulting $p$-values were lower than 0.005, demonstrating that our method achieves significantly improved performance despite the conservative threshold (see Table G). Furthermore, to assess the effectiveness of each major component proposed in this paper, we performed two-sided paired $t$-tests between the full model and its ablated variants. It indicated that all components were found to be statistically significant, highlighting in particular the effectiveness of our template-free Gaussian deformation module.

| Method | vs GART (Lei et al., 2024) | vs GaussianAvatar (Hu et al., 2024a) | vs 3DGS-Avatar (Qian et al., 2024a) | vs ExAvatar (Moon et al., 2024) | vs w/o VE | vs w/o SMAD |
|---|---|---|---|---|---|---|
| $p$-value | $5.6 \times 10^{-6}$ | $4.2 \times 10^{-6}$ | $6.9 \times 10^{-5}$ | $5.7 \times 10^{-5}$ | $3.3 \times 10^{-4}$ | $1.7 \times 10^{-4}$ |

**Table G:** Statistical significance ($p < 0.05$). We performed a two-sided paired $t$-test against each baseline method conservatively at $p = 0.05$ to relieve a multi-comparison issue. Our method exhibit statistical significance compared to the baselines and ablated ones, suggesting that our full method have significant performance improvement.

## D.3 GENERALIZATION

We further evaluate the generalization performance. Fig. J shows the distribution of training poses and test poses on a t-SNE plot, as well as the performance on the in-the-wild dataset. The blue box indicates the distribution of the training motion, and the orange box indicates that of the test motion. Even though the test motion was unseen during training, our method demonstrates improved generalization performance. In Fig. J, we plot how the perceptual error changes relative to the motion similarity between the training and testing data, measured via normalized cross-correlation (NCC) between the time-varying 3D conditional poses. We observe a more pronounced increase in the error for the baseline as the testing motions deviate further from the training data. Furthermore, while it shows a large variation in standard error that increases as motion similarity decreases, our method consistently maintains a low level of standard error. Since the standard error is computed between the train and test motions, a lower value indicates less overfitting and suggests better generalization performance. This suggests that our method exhibits robust generalization performance on par with the linear skinning model of the conventional template parametric model.

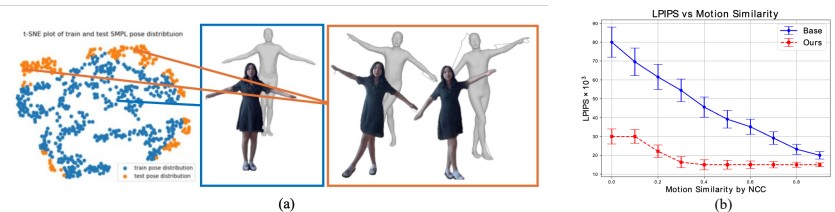

**Figure J:** (a) Visual check of in-distribution (*blue*) and out-of-distribution (*orange*) driving poses with t-sne plot. (b) Average perceptual metric (LPIPS; lower is better) with standard error plot of 4D-Dress over motion similarity between train and test set. Our method (*red*) maintains consistent rendering performance even for test motions with low similarity to the training motion—showing relatively less performance degradation compared to high-similarity cases—whereas a baseline (*blue*) exhibits a significant drop in perceptual quality when handling test motions with low motion similarity.

## E    MORE RESULTS

We present additional qualitative comparison results in the remainder. Please refer to the Fig. K, L, M, N, O, P.

## F    LIMITATIONS

First, our method struggles under dynamics that involve sudden and large motion changes. Although we designed the system to account for temporal context using velocity encoding and auto-regressive modeling, it still has difficulty predicting the emergence of node accelerations that lie outside the training distribution. Second, our method does not model multi-garment interactions. The current Gaussian graph is a single-layer structure that captures the overall clothed shape and ensures deformations that preserve this global structure. However, it does not model interactions between garments or predict their independent motions. In future work, we aim to address these limitations. To tackle the first challenge, we could introduce a deformation model that considers bidirectional temporal context. Alternatively, incorporating a generative flow matching technique that predicts a bundle of vectors (a vector field) may offer a promising way to learn and represent the distribution of complex motions. To address the second issue, we could consider constructing a hierarchical Gaussian graph and introducing a graph neural network to model interactions between different garments. However, achieving this would require highly accurate semantic segmentation between garments. Currently, such segmentation remains difficult in the presence of diverse self-occlusions and depth ambiguities from a single-view video. Therefore, enabling high-quality multi-garment segmentation from a single video alone would itself be a highly challenging yet exciting direction for future research in hierarchical Gaussian deformation modeling.

## G    BROADER IMPACTS

**Potential Negative Societal Impacts.** Our technology could be misused, leading to negative societal consequences. One major risk is Deepfake-style impersonation: a realistic avatar of a person could be created without consent and used to impersonate them, enabling misinformation or fraud. The ability to replicate someone's likeness from a single video also raises privacy concerns, as individuals could have their image replicated and misused in unwanted ways, which can erode trust in digital media. It could also impact creative industries: unauthorized digital replicas of actors might violate intellectual property rights and undermine the entertainment industry's economy, and a proliferation of lifelike fake characters could confuse audiences and devalue genuine performances. These risks underscore the need for ethical guidelines and safeguards to prevent malicious use of AI-driven avatar technology.

**Broader Impact.** Our work offers positive implications for research, industry, and consumers. *Research Community:* Our method introduces a new approach to animatable avatars using 3D Gaussian Splatting, advancing neural rendering, and provides an in-the-wild dynamic clothing dataset to spur further research on neural avatars and secondary motion modeling.
*Industry:* The improved realism and efficiency of our approach can benefit digital human applications in entertainment, gaming, and virtual reality by enabling creators to produce lifelike characters with realistic cloth dynamics from minimal input, allowing immersive real-time experiences.
*Consumers:* More realistic and animatable avatars mean more immersive virtual experiences for end-users. Users in VR and gaming will be able to interact through avatars that mirror their appearance and clothing motion, enhancing their sense of presence. By bridging real and virtual representations, our work enriches virtual experiences.

**Ethics Statement.**    This work makes use of both publicly available datasets (e.g., ZJU-MoCap, 4D-Dress) and a newly collected dataset, LoCo-Human, which contains in-the-wild monocular video sequences of clothed human subjects. For all publicly available datasets, we adhere to their respective license terms and usage conditions. For LoCo-Human, all participants were recruited with explicit informed consent, covering video recording, research use, and potential public release of the anonymized dataset. No minors or vulnerable populations were included. Personally identifying metadata beyond facial and body appearance was not collected, and access to raw recordings will be restricted. The dataset will be released after peer review with a research-only license

prohibiting redistribution and commercial use, and with clear take-down procedures if requested by participants. We acknowledge that technologies enabling high-fidelity 3D avatar reconstruction from monocular videos may be misused for malicious purposes (e.g., impersonation, non-consensual content generation). To mitigate such risks, we emphasize responsible use of the dataset and models, encourage watermarking or detection mechanisms for synthetic outputs, and restrict the release of model weights to verified research purposes only. We also recognize the possibility of bias due to the limited diversity of clothing types, subjects, and motions in LoCo-Human. We report dataset composition and limitations transparently and encourage future work to expand demographic and cultural coverage for fairness and inclusivity. No sensitive medical or financial information is used in this work. Institutional review board (IRB) approval was not required, but ethical considerations regarding informed consent, privacy, and responsible release were carefully followed.

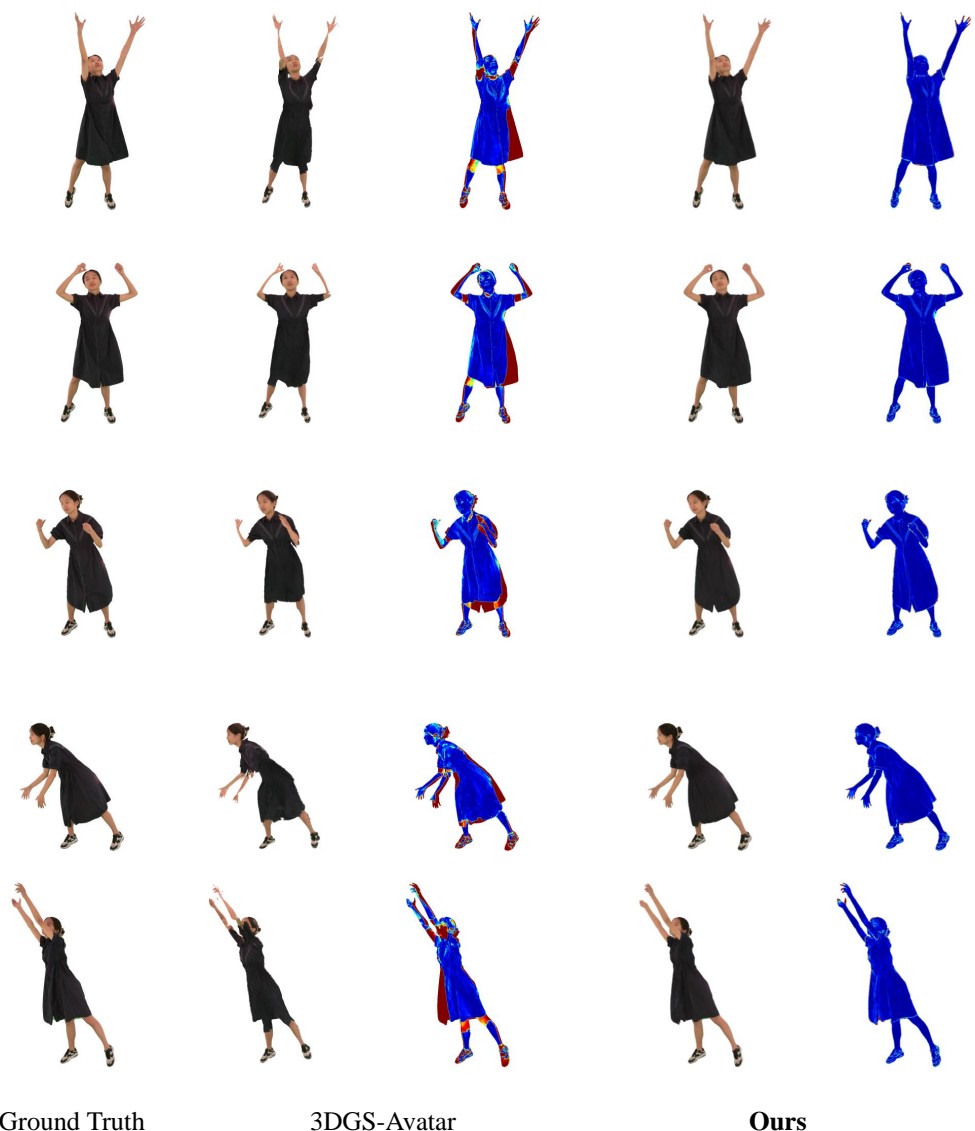

Ground Truth          3DGS-Avatar          **Ours**

**Figure K:** Qualitative Results of *00148* subjects on 4D-Dress dataset, compared to Qian et al. (2024b) with multiple motions across time axis.

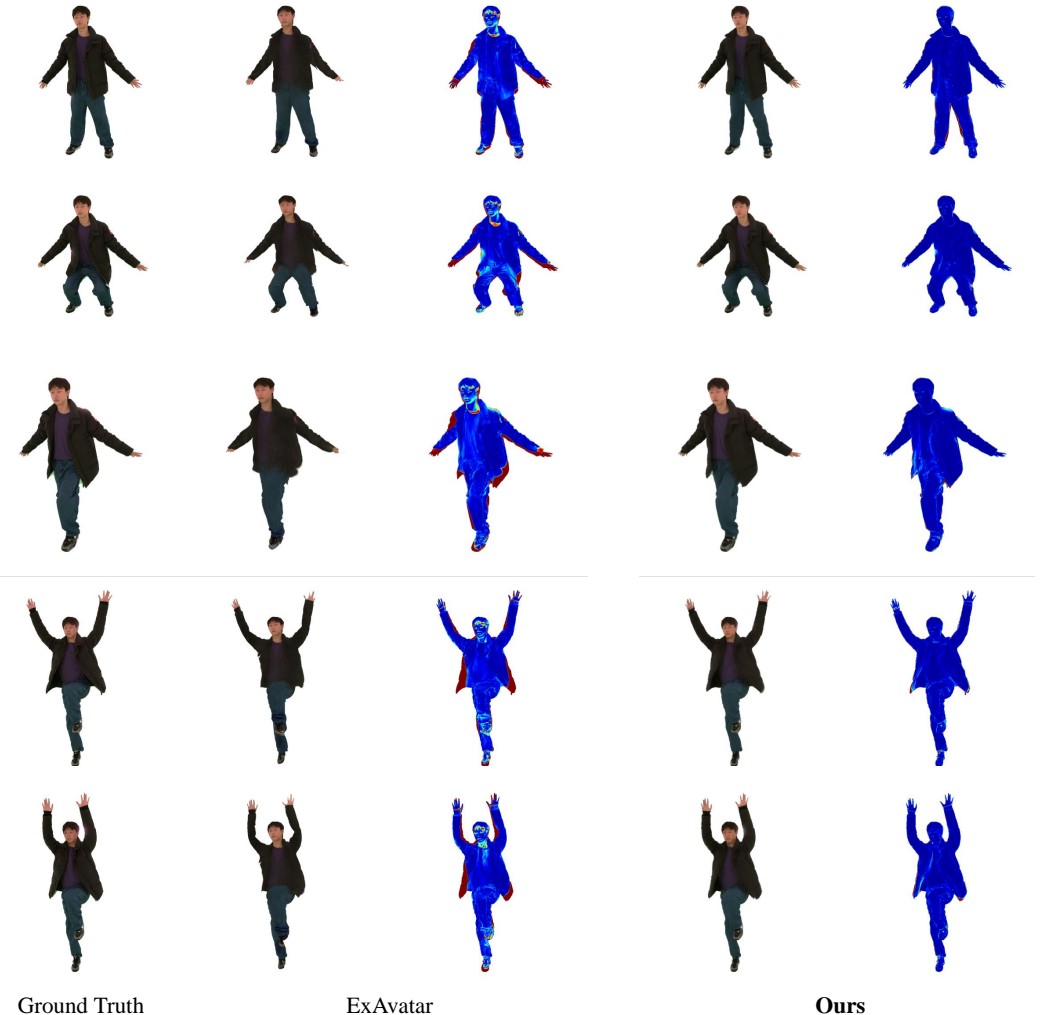

Ground Truth                    ExAvatar                    **Ours**

**Figure L:** Qualitative Results of *00169* subjects on 4D-Dress dataset, compared to Moon et al. (2024) with multiple motions across time axis.

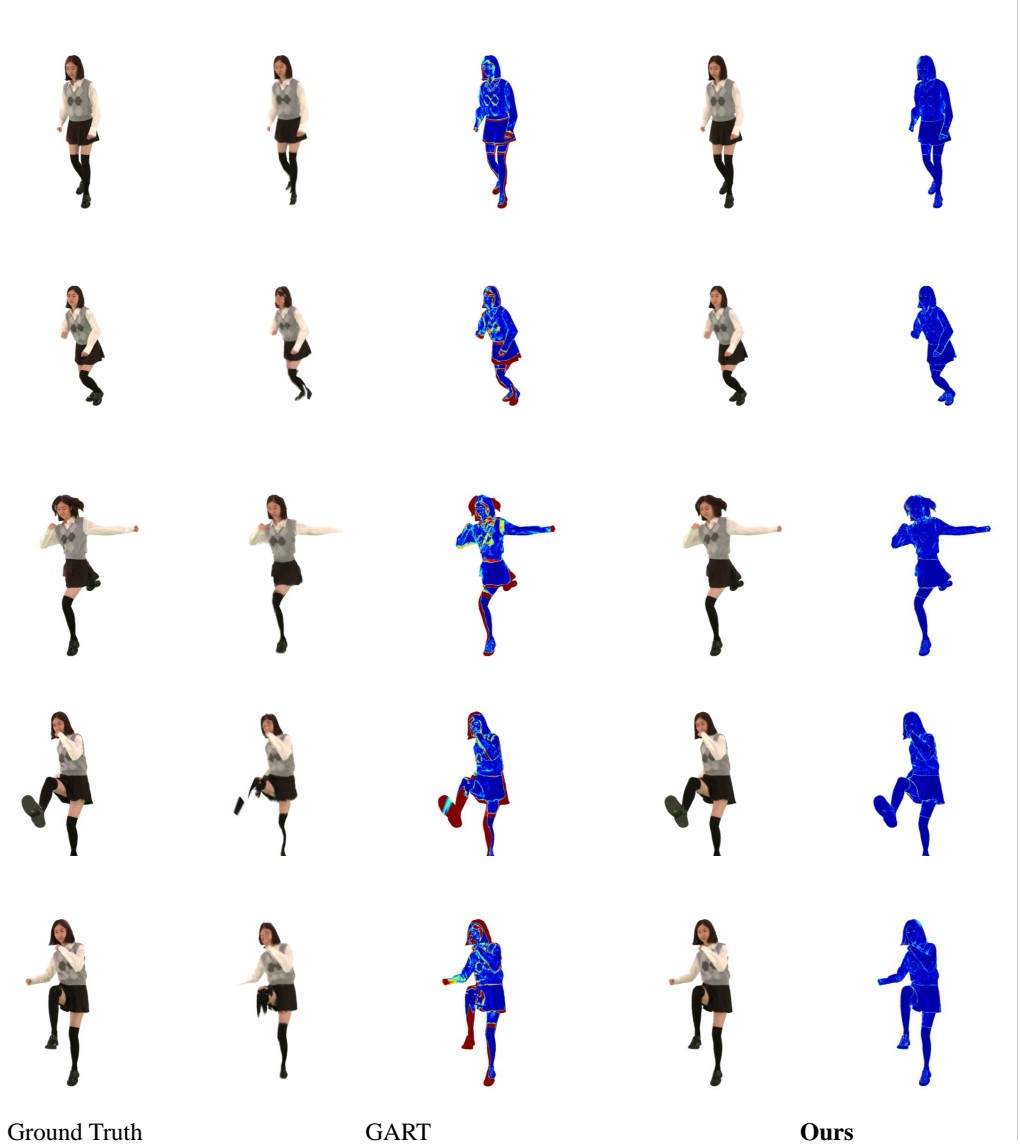

Ground Truth                     GART                     **Ours**

**Figure M:** Qualitative Results of *00170* subjects on 4D-Dress dataset, compared to Lei et al. (2024) with multiple motions across time axis.

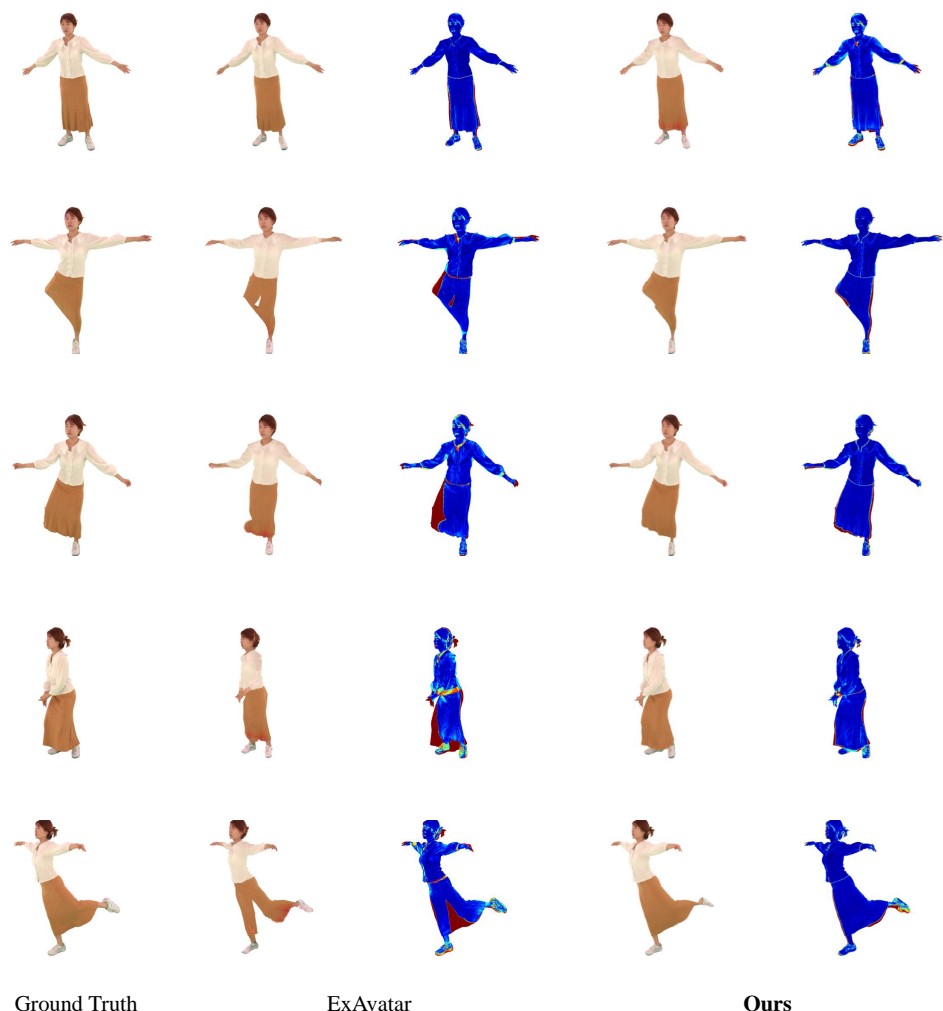

Ground Truth        ExAvatar        **Ours**

**Figure N:** Qualitative Results of *00185* subjects on 4D-Dress dataset, compared to Moon et al. (2024) with multiple motions across time axis.

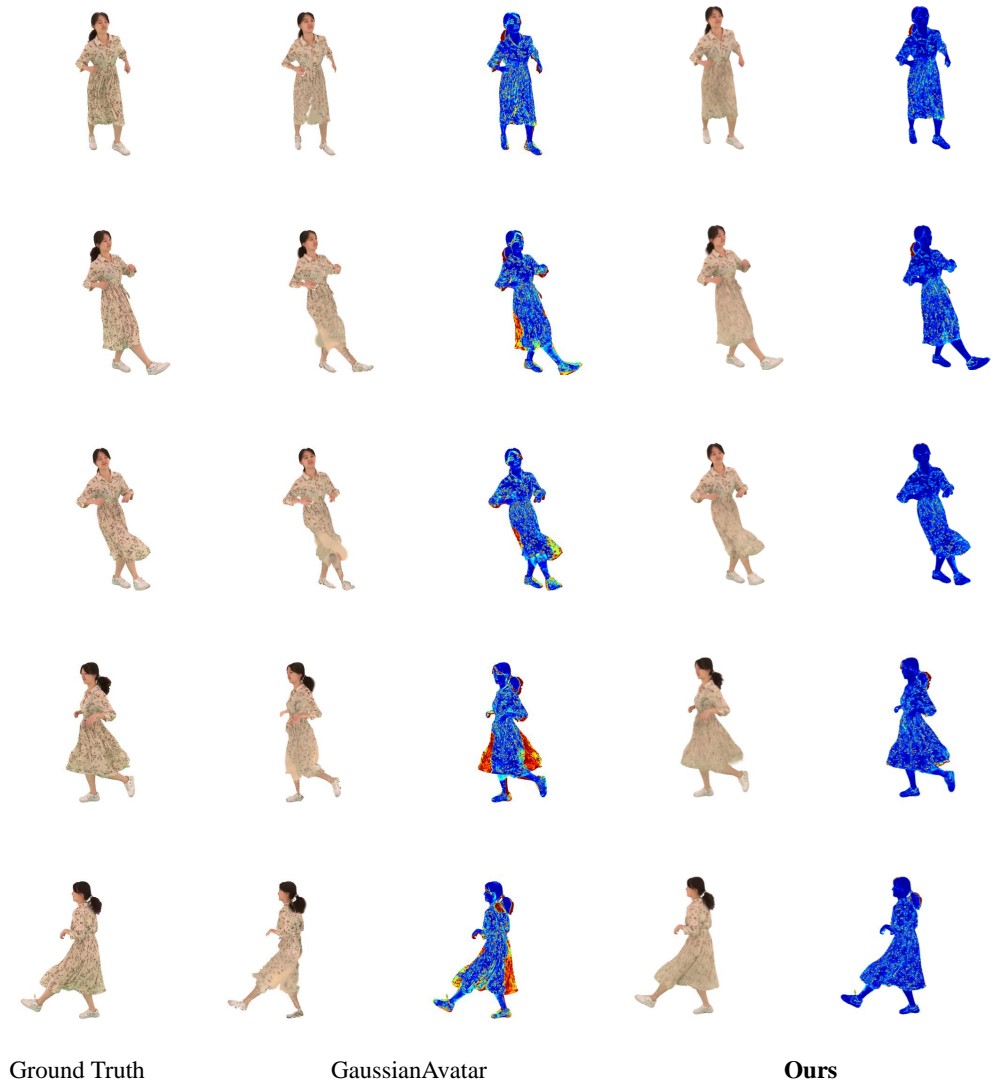

Ground Truth  GaussianAvatar  **Ours**

**Figure O:** Qualitative Results of *00187* subjects on 4D-Dress dataset, compared to Hu et al. (2024a) with multiple motions across time axis.

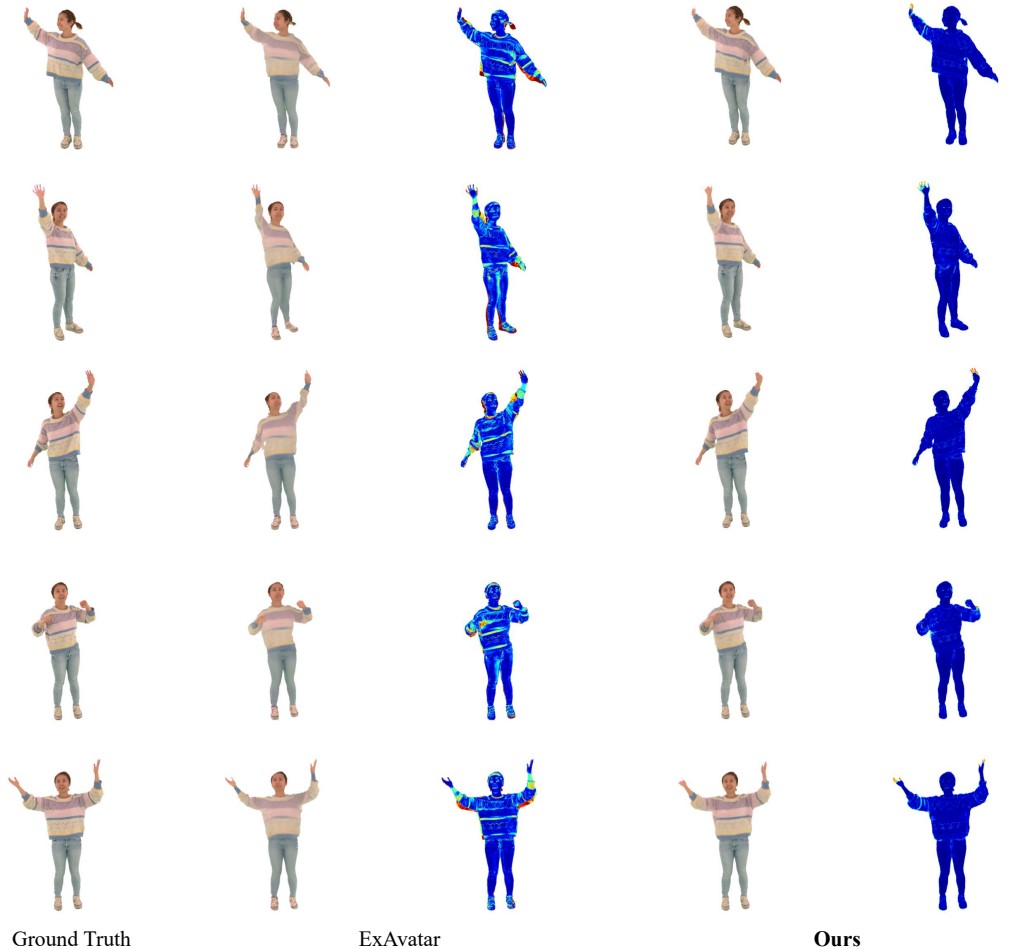

Ground Truth      ExAvatar      **Ours**

**Figure P:** Qualitative Results of *00190* subjects on 4D-Dress dataset, compared to Moon et al. (2024) with multiple motions across time axis.

