# OpenReview forum: "SMAGA: Secondary Motion-Aware 3D Clothed Gaussian Avatars from Monocular Videos"
_ICLR.cc/2026/Conference — ICLR 2026 Poster_

### Official Review · Reviewer_ZfFP · 2025-10-30

**Soundness:** 3
**Presentation:** 3
**Contribution:** 3
**Rating:** 6
**Confidence:** 4

**Summary:**

This paper presents a novel, two-stage framework for creating animatable 3D Gaussian Splatting (3DGS) avatars from a single monocular video, with a specific focus on realistically modeling the "secondary motion" of loose-fitting garments. The authors identify two key failings of prior work: (1) poor initialization of Gaussians due to a mismatch between "naked body" templates and clothed geometry, and (2) an inability to model complex cloth dynamics due to deformation models that lack temporal context.

To solve this, this paper proposes:
1)Personalized Gaussian Initialization (PGI): A pre-processing stage that first trains a 4D NeRF to create a canonical, clothed representation of the subject, from which the initial 3D Gaussians are extracted. This ensures the Gaussians' starting positions match the actual clothed shape.

2)Secondary Motion-Aware Deformation (SMAD): A novel deformation model that represents the canonical Gaussians as a graph. A GNN then autoregressively predicts the position and velocity of the graph nodes, inspired by a second-order mass-spring-damper system. By explicitly encoding a buffer of past velocities, this model captures the temporal context crucial for realistic cloth motion.

This paper validates the method against several recent 3DGS-avatar baselines on three datasets, including the new "LoCo-Human" in-the-wild dataset they introduce. The experiments show state-of-the-art results, with quantitative Table 1 and qualitative Fig. 3, 4  data demonstrating a clear superiority in modeling loose clothing and avoiding common artifacts like skirt-splitting.

**Strengths:**

Clear Problem Definition: The paper does an excellent job of identifying, diagnosing, and illustrating a significant weakness in current monocular avatar creation: the failure to model secondary motion for loose clothing. The analysis of initialization mismatch and temporal-unaware deformation is insightful and directly motivates the proposed solution.

Novel and Sound Methodology: The PGI stage is a clever solution to the initialization problem. Using a deformable NeRF to build a subject-specific, clothed canonical space is a much more robust approach than trying to fit a generic, naked template (e.g., SMPL) to loosely-clothed subjects.

The SMAD module is the paper's core strength. Moving from a simple pose-conditioned deformation to a physics-inspired, autoregressive GNN is a significant and logical step. The "Velocity Encoding" (VE), which incorporates a buffer of past motions, is a direct and effective way to model the history-dependent nature of cloth dynamics.

Extensive and Convincing Experiments: The experimental validation is a major strength. The method is compared against multiple strong, recent 3DGS-based methods. Evaluation spans three distinct challenges: novel view synthesis (ZJU-MoCap), novel pose synthesis (4D-Dress), and in-the-wild generalization (LoCo-Human). The method achieves state-of-the-art quantitative results across the board (Table 1). The qualitative results (Fig. 1, 3, 4) are particularly compelling, clearly showing the elimination of artifacts (like skirts splitting or "needle" artifacts) that plague other methods.

Dataset Contribution: The introduction of the LoCo-Human dataset, featuring in-the-wild videos of subjects in loose clothing, is a valuable contribution to the community, which lacks such data.

**Weaknesses:**

Insufficient Ablation Experiments: There is no ablation experiments and no discussion about the selection of GNN-based autoregressive deformer with other models.

Missing Experiment Details: In the ablation study of Physics & Graph Design, there is no detailed explanation about the ablation content. For example, physics in A0 and Contact-aware cross-edges in A3 are ambiguous.

Descriptive Ambiguity:  In the experiment part, this paper describes that the LoCo-Human features five Loose-Clothed Humans performing 5 dynamic and 1 static motions per subject. However, in the data statistics of the Appendix, this paper states that the dataset comprises 5 unique subjects, each recorded across 5 sequences.

Cost of PGI Stage: The PGI stage relies on training a full 4D NeRF for each subject before training the main SMAD model. Table F shows that this stage takes 12.5 hours, nearly 3x longer than the 4.5-hour SMAD training. While the fast inference (26 fps)  is excellent, the total training time (17 hours) is substantial. This high "personalization" cost should be more clearly discussed as a trade-off.

Figure Quality Issue: Some texts in the figure are not correctly ordered, such as the text “Motion” in Fig. 1 (b). There is a redundant line on the right of Figure L.

Others: In the More Results part of the Appendix, the citation of the figure is not correct.

**Questions:**

Why does this paper choose GNN-based? Is there any model selection ablation experiment?

What do the physics in the ablation study represent? Are they the two physics-based losses in Equations 11 and 12?

And what do the hierarchical body–cloth graph and contact-aware cross-edges represent?

---

> ### Author Response · Authors · 2025-11-27
>
> We sincerely thank the reviewer for their thoughtful and encouraging feedback. Below, we provide detailed responses to the reviewer’s valuable questions and suggestions.
>
> ---
>
> ### **Response to W1&Q1: “Insufficient ablations on the choice of a GNN deformer.”**
>
> Thank you for the insightful comments. To address this concern, we additionally conducted a controlled comparison against a carefully designed MLP-based autoregressive deformer, following the architecture used in *Zheng et al., 2021*. This baseline uses the same inputs (positions, encoded velocities) as our method, ensuring a fair comparison.
>
> | Model       | PSNR ↑ (Test) | SSIM ↑ (Test) | LPIPS ↓ (Test) | PSNR ↑ (Train) | SSIM ↑ (Train) | LPIPS ↓ (Train) |
> | ----------- | -------------- | -------------- | --------------- | ------------- | ------------- | -------------- |
> | MLP         | 25.46          | 0.954          | 0.056           | 27.97         | 0.973         | 0.044          |
> | **Ours**    | **27.89**      | **0.963**      | **0.040**       | **28.64**     | **0.984**     | **0.037**      |
>
> On the 4D-Dress subjects, we evaluate both models on (i) motions seen during training and (ii) held-out unseen motions. As summarized in the above Table, the MLP deformer fits the training motion but degrades significantly on unseen motion, while the GNN-based deformer remains substantially more stable and accurate. This confirms that the graph-based formulation provides stronger structural priors and better generalization for clothed-human deformation.
>
> We include this ablation in Sec.4.3.
>
> ---
>
> ### **Response to W2&Q2&Q3: "Missing experimental details of physics & graph design"**
>
> Thank you for pointing this out. In the context, “Physics” refers primarily to the two physics-inspired regularizations in Eq. (11) (isometry preservation) and Eq. (12) (damping), which impose length-preserving deformation and suppress high-frequency velocity noise.
>
> A0 represents base configuration of vanilla GNN with auto-regressive deformation of predicting positions and velocities with finitie-difference method, without any physically plausible regularizations.
>
> A2 (hierarchical graph) represents the introduction of learnable spring stiffness coefficients that adaptively separates rigid body regions and highly deformable cloth regions. This helps maintain stability in body-proximal Gaussians while allowing flexible, larger-amplitude deformation in garment areas.
>
> A3 (contact-aware edges) represents the introduction of message-passing strategy that encodes not only node features, but also edge features to understand explicit interactions between neighboring nodes.
>
> For clarity, we revise the notation of A0-4 methods in Table 2, and described it in details in Sec.4.3.
>
> ---
>
> ### **Response to W3: “Descriptive ambiguity in the LoCo-Human dataset.”**
>
> Thank you for the comment. The discrepancy arose because the main text describes the dataset by motion type, while the Appendix reports it by evaluation sequences. Concretely, each subject performs 1 static motion and 5 dynamic motions, totaling 6 sequences per each subject. We have rewritten the dataset description to remove this ambiguity and clearly state the description in the Appendix.
>
> ---
>
> ### **Response to W4: “Cost of PGI Stage and trade-off.”**
>
> Thank you for highlighting this point and for the constructive suggestion. Following the comment, we conducted an additional experiment to explicitly examine the performance trade-off with respect to PGI training cost. Specifically, we varied the PGI training time such that the total training time of our pipeline becomes [8, 12, 16, 20, 24, 28] hours, and measured the resulting PSNR performance. For comparison, we also evaluated a competing state-of-the-art method under varying training budgets. Please refer to Fig.7 and Sec.4.3 in the revised manuscript.
>
> The results clearly show that, unlike prior methods whose limited model capacity leads to marginal or saturated improvements under longer training, our method consistently benefits from additional PGI optimization. In particular, models with longer PGI training exhibit noticeable gains in reconstruction fidelity and temporal stability. This indicates that our pipeline has a higher effective capacity and is capable of leveraging extended training to achieve substantially improved dynamic appearance modeling. At the same time, the new results demonstrate that this additional cost directly translates into measurable performance benefits—particularly in faithfully capturing loose-clothing dynamics—which prior methods cannot obtain even with comparable training budgets.
>
> ---
>
> ### **Response to W5&W6: "Formatting Issues"**
>
> Thank you for pointing out these formatting issues. We have corrected the misaligned text in Fig. 1(b), removed the redundant line in Figure L, and fixed the incorrect figure citations in the Appendix. All affected figures and references are updated in the revised manuscript.
>
> ---

---

### Official Review · Reviewer_Gn6q · 2025-10-30

**Soundness:** 3
**Presentation:** 3
**Contribution:** 3
**Rating:** 6
**Confidence:** 4

**Summary:**

The paper proposes a framework for modeling dynamic appearances of avatars from single monocular videos using 3DGS, with an emphasis on loose-fitting garments and secondary motion (e.g., skirt flutter). Instead of relying on pre-defined template-based initialization and skeletal skinning-based animation, this paper obtains dense Gaussian initialization through personalized 4D NeRFs, constructs a velocity-encoded Gaussian graph and finally learns a secondary motion-aware deformation  module that autoregressively predicts second-order dynamics. The authors also collected a new LoCo-Human dataset, containing in-the-wild videos with dynamic cloth motion. Experiments show strong improvements over state-of-the-art 3DGS-based avatar methods (e.g., 3DGS-Avatar, ExAvatar) on multiple datasets.

**Strengths:**

* The velocity-encoded Gaussian graph is a plausible design: it introduces physical intuition (mass–spring–damper) while maintaining differentiability for network learning. Such a design addresses a key gap in existing 3DGS-based avatars that only model the per-frame pose-to-deformation mapping and neglect the second-order dynamics.

* The LoCo-Human dataset addresses the lack of dynamic loose clothing under monocular capture in existing benchmarks. The dataset ethics statement is detailed, with informed consent and consideration for potential misuse.

* The paper is generally well-written and easy to follow. Figures are clear, with helpful comparisons and ablations.

**Weaknesses:**

* Constructing a node graph for modeling loose garments is not a new thing. Similar ideas have been well explored in previous methods using mesh-based representations like "Real-time Deep Dynamic Characters" (Habermann et al 2021). More discussions on the relationship between this paper and existing methods are necessary.

* Although the paper draws analogies to second-order mass–spring systems, the GNN-based updates are learned implicitly, and no physical consistency (e.g., mass, stiffness calibration) is enforced. The “physics-inspired” term might overstate the grounding; clarification on whether parameters (k_ij, γ_i) are learned or derived would help.

**Questions:**

Missing citation:

Li et al. Animatable Gaussians: Learning Pose-dependent Gaussian Maps for High-fidelity Human Avatar Modeling. CVPR 2024

---

> ### Author Response · Authors · 2025-11-27
>
> We sincerely appreciate the reviewer for their thoughtful feedback. We have carefully addressed all the comments provided.
>
> ---
>
> ### **Response to Weakness: “Graph construction for loose garments is not new (e.g., Real-time Deep Dynamic Characters).”**
>
> Our contributions differ in line of the previous works:
>
> 1. Different level of supervision.: Prior methods build graphs on ground-truth clothed meshes with multi-view or simulation supervision. In contrast, we propose a method for constructing a graph from unstructured Gaussian primitives, together with a person-specific Gaussian initialization, without meshes or multi-view geometry.
>
> 2. Template-free clothed avatar deformation.: The previous works depend on pre-defined subject-specific body/garment templates and 4D ground-truth scans. In contrast, we propose template-free approach for dynamics of clothed avatars, introducing secondary motion-aware deformation with auto-regressive prediction of physical quantities (positions and velocities) with finitie-difference method.
>
> 3. New combination with 3DGS. To our knowledge, no prior work presents an integrated pipeline of structuring Gaussian primitives with second-order dynamics with autoregressive GNN.
>
> We update the Related Work to clarify our distinctions in the revised manuscript.
>
> ---
>
> ### **Response to Weakness: “'Physics-inspired' may overstate the grounding; clarification on k_ij, gamma_i would help.”**
>
> We appreciate this comment and agree that our method should not be interpreted as a fully physically consistent simulator. Our use of the term “physics-inspired” refers specifically to the second-order state formulation and kinematic quantities (positions and finite-difference velocities), not to calibrated physical parameters.
>
> Concretely, $k_{ij}$ and $\gamma_{i}$ are not explicitly instantiated or constrained in our implementation. Instead, the GNN learns interaction patterns that resemble spring and damping behavior through message passing, rather than derived from known material constants. Our goal is therefore to achieve temporally coherent, inertia-like cloth motion from a single video, not to recover ground-truth physical properties such as material parameters.
>
> We clarify that the $k_{ij}$ are implicitly learned, “physics-inspired” is meant in the standard hybrid-modeling sense of using physical state variables and a second-order update structure rather than implying strict physical correctness. We update corresponding sentences in Sec.3.1 and Sec.4.3.
>
> ---
>
> ### **Missing Citation**
>
> Thank you for pointing this out. Animatable Gaussians (Li et al., CVPR 2024) is indeed relevant, and we cite it in the revised manuscript.
>
> ---

---

### Official Review · Reviewer_1tPA · 2025-10-30

**Soundness:** 3
**Presentation:** 3
**Contribution:** 2
**Rating:** 6
**Confidence:** 4

**Summary:**

* This paper introduces a method for creating a person-specific animatable avatar from a single video.
* Method combines a commonly used approach of attaching gaussian splats to a human LBS model, and intorduces a secondary motion modeling element on top. Namely, the gaussian location and its moments are modeled as a dynamic system, parameterized by an autoregressive graph neural network conditioned on per-frame latent codes and representations of history/neighbors states and velocity.
* Evaluation is conducted on a set of standard benchmarks (ZJU-Mocap, 4D-Dress), as well as a newly introduced dataset, comprised of videos more suitable for evaluation on subjects with loose garments (LoCo-Human).

**Strengths:**

*Clarity/quality:*
- Paper is relatively well-written and is easy to follow.
- Method seems easy to implement.

*Originality / significance:*
- Existing approaches for avatar modeling indeed lack realistic 2ndary
motion modeling, and modeling it with a hybrid physically inspired approach
(predicting physical properties with a NN) sounds like a technically sound
approach.

*Evaluation:*
- Quantitative and qualitative comparison indicates that the method
performs favorably compared to the chosen baselines.
- On the examples shown in the supp video clothing deformations indeed
look convincing.

**Weaknesses:**

*Method Limitations:*
- Method is person-specific, which means that a new model is trained per input video, and if the information (e.g. about the back of the body) is missing, there is no way to recover it from a prior.
- Similarly, one can be sceptical that complex physics of clothing
eformations can be learned from a single video without relying on any data-driven prior or a large dataset.
- (Arguable) This means that the method is unlikely to generate truly realistic motions for poses outside of training distribution, and is likely
overfitting to training sequences.


*Novelty / Significance:*
- The overall GS+LBS pipeline is not novel, not fully clear if the proposed dynamics formulation combined with GNN has
been done before.

*Experimental Evaluation:*
- It is unclear why the methods are different across different bencharks.
- Not fully related to the papers itself, but the quality of ZJU-Mocap
dataset is extremely poor to the point that results on that dataset
are not informative.
- (Minor) For ablation study, it would be useful to understand how the A0 performs compared to baselines (is it already better?). Also, was it conducted on a single subject? If so, it is unclear if the results
are very reliable.
- (Minor) It is actually unclear if the method is in any way specific to a single video setup (arguably, it is not - see method limitations), and there exists a variety of datasets of much higher quality (ActorsHQ, Goliath) which could inform whether the formulation provides
extra benefits in less noisy scenarios (and enable using more
baselines).

**Questions:**

1. Autoregressive formulation is known to be prone to error accumulation issues. Have authors considered evaluating their method on longer sequences
to confirm no "explosions" happens due to this?

2. Why different methods across different benchmarks?

---

> ### Author Response · Authors · 2025-11-27
>
> We sincerely thank the reviewer for the thoughtful and encouraging feedback. Below, we provide careful and respectful responses to the concerns you have thoughtfully raised.
>
> ---
>
> ### **Method Limitations**
>
> **Person-specific & lack of prior**
>
> Our method is intentionally designed for the single-video, per-subject setting, which is the standard paradigm in monocular avatar methods such as HumanNeRF, GaussianAvatar, 3DGS-Avatar, ExAvatar, etc. Our goal is to recover high-fidelity secondary motion from a single sequence rather than build a universal prior.
>
> ---
>
> **Generalization**
>
> We evaluated generalization quantitatively.
>
> * On motion-space similarity (Fig. J): Even when the cosine similarity between training/test motions is low, our model maintains consistent perceptual quality, unlike baselines. t-SNE distributions and unseen-motion samples confirm stable deformation outside the training cluster.
> * Train–test statistics: On 4D-Dress, we performed a paired t-test between train and unseen test sequences, yielding p-val = 0.37, showing no significant degradation. It suggests that the model does not overfit the observed frames.
> * Reason: Our second-order autoregressive update with finite-difference velocities provides a physically meaningful state, known to improve rollout stability and mitigate overfitting in non-rigid dynamics.
>
> We incorporate the additional experiments and analyses in Sec.4.3.
>
> ---
>
> **Learning complex cloth physics from a single video**
>
> We appreciate the reviewer for the insightful feedback. We agree that full physical simulation of garments from monocular input is inherently under-constrained, and we do not claim to recover physically accurate material parameters or detailed physical interactions from a single sequence. Instead, our goal is more modest and more aligned with the single-video avatar literature: to learn perceptually plausible, temporally coherent secondary motions, without requiring population-level data-driven priors.
>
> ---
>
> ### **Novelty**
>
> Our pipeline is not GS+LBS. A key contribution is removing template articulation, which fundamentally limits loose-cloth modeling. Our novelty lies in:
>
> 1. A template-free, velocity-encoded Gaussian graph constructed directly on canonical clothed Gaussians.
> 2. An autoregressive second-order GNN that predicts Gaussian deformation through physical-state quantities (positions + finite-difference velocities).
>
> To our knowledge, this combination has not been explored in monocular 3DGS-avatar literature and results in more stable and realistic secondary motion.
>
> ---
>
> ### **Experimental Evaluation**
>
> **Different baselines across datasets**
>
> This follows availability, not selective choice. ZJU-MoCap is the standard benchmark but does not include loose garments. We therefore additionally evaluate on 4D-Dress and LoCo-Human, and include baselines that (i) offer official inference code for custom subjects and (ii) explicitly model non-rigid dynamics in 3DGS settings.
>
> ---
>
> **On ZJU-MoCap**
>
> We agree it is limited for evaluating our contributions. It is included only for completeness; our core claims are demonstrated on datasets with loose clothing and dynamic motions.
>
> ---
>
> **Single-video setup**
> Thank you for raising this point. Our method is intentionally designed for the single-video setting and evaluated on datasets featuring loose-fitting garments and dynamic motions. Although ActorsHQ and Goliath are high-quality datasets, they are captured under dense, calibrated multi-view studio setups, making them more suitable for methods that leverage multi-view geometry and supervision. This setting is fundamentally misaligned with our causal monocular formulation, which targets loose-fitting clothing dynamics from a single video.
>
> ---
>
> ## **Error Accumulation**
>
> Thank you for the interesting suggestion. To analyze and reflect on this point, we captured two types of motion sequences, each lasting over 30 seconds: (a) dynamic pose sequence, and (b) a repetitive pose sequence. We evaluated our proposed method, and also conducted a comparative analysis with and without our proposed velocity encoding scheme to evaluate its impact. Note: Metrics are reported in the order of PSNR / SSIM / LPIPS for each cell.
>
> |  | (a) Dynamic Pose | (b) Repetitive Pose |
> | --- | --- | --- |
> | w/o V.E | 24.47 / 0.949 / 0.050 | 24.69 / 0.950 / 0.049 |
> | **w/ V.E (Ours)** | **25.65 / 0.955 / 0.044** | **26.84 / 0.960 / 0.039** |
>
> These results demonstrate that our velocity encoding scheme consistently improves performance on long-sequence motions, with particularly notable gains in the repetitive pose scenario. Our velocity encoding scheme appears to mitigate this issue by incorporating a history of multiple past states, rather than relying solely on the most recent estimate. This allows the model to remain robust even when the immediate past prediction is noisy, reducing the risk of cumulative drift. In Sec.4.3, we include the experiements and discussions in detail.
>
> ---

---

### Official Review · Reviewer_Wxo4 · 2025-11-01

**Soundness:** 3
**Presentation:** 4
**Contribution:** 3
**Rating:** 8
**Confidence:** 4

**Summary:**

This paper tackles the challenging problem of creating animatable 3D human avatars from a single monocular video, with a specific focus on realistically modeling the secondary motion of loose-fitting clothing (e.g., skirts, coats). The authors identify two primary failings of current 3D Gaussian Splatting based methods: cloth shape-agnostic initialization and temporal context-unaware deformation. To solve this, the paper proposes a two-stage framework. The first is Personalized Gaussian Initialization (PGI) that avoids naked-body templates by first training a deformable 4D NeRF on the input video. A set of canonical 3D Gaussians is then extracted from this person-specific clothed shape. The second is Secondary Motion-Aware Deformation (SMAD) where the canonical Gaussians are structured into a graph. A GNN-based deformer then learns to animate these Gaussians.

The authors also introduce a new in-the-wild dataset, LoCo-Human, featuring subjects in loose clothing captured with smartphones. Experiments on LoCo-Human and other benchmarks (4D-Dress, ZJU-MoCap) show that the proposed method outperforms recent 3DGS-avatar baselines, particularly in the quality and realism of cloth animation.

**Strengths:**

1. The authors identify and illustrating (with Fig. 1) a key failure mode of modern animatable avatar methods, i.e., modeling varying clothing dynamics with time. This is in geenral a notoriously difficult and important problem, and the authors' analysis of why current methods fail (template mismatch and lack of temporal context ) is convincing. I would suggest adding some other relevant works [1,2,3,4] for trying to tackle the goal of modeling loose clothing with/without a template using point clouds and [5,6] that similarly use implicit representations for creating an initial representation to model clothing. [7] PhysGaussian is one of the works using physics to model dynamic Gaussians.
2. The proposed two-stage solution directly addresses the identified problems. The PGI stage creates a high-fidelity canonical representation of the clothed individual, which is a much better starting point for deformation. Note that this is explored in various other works as well [5,6]. The SMAD module's design is the paper's main strength. Moving from a per-frame, pose-conditioned model to an autoregressive, physics-inspired one is an important shift. Using velocity encoding to give the GNN a temporal state allows it to learn complex dynamics (like inertia and damping), which is something static models cannot do.
3. The experiments are thorough and are done on standard benchmark (ZJU-MoCap) as well as on more challenging and relevant datasets (4D-Dress, LoCo-Human) that contain subjects wearing loose clothing. The quantitative results (Table 1)  show improvement over all baselines across all three datasets. The qualitative results (Fig. 3, 4) visibly demonstrate the method's ability to avoid common artifacts like skirt-splitting and needle artifacts seen in competing work.
4. The ablation studies (Table 2, Fig. 6) effectively isolate the contributions of the key components, showing that Velocity Encoding and the graph-based SMAD deformer are essential to the performance gains.
5. The proposed LoCo-Human dataset is a valuable contribution. As it is captured in-the-wild with standard smartphones, it lowers the barrier to entry and will likely spur further research in this area.

**Weaknesses:**

1. The paper explains that N canonical Gaussians are down-sampled to M graph nodes (M << N) and that the GNN deforms these M nodes. However, it never explains how the deformation of these M nodes is propagated back to the full N Gaussians for the final rendering. Is each Gaussian rigidly attached to the nearest graph node? Is there an interpolation scheme (e.g., barycentric, LBS-like)? This is a critical, missing link in the pipeline.
2. The title is "Dynamic Texture Modeling...". However, the paper's novelty and focus are on dynamic geometry and motion. The authors model the deformation of 3D Gaussians (position, covariance) autoregressively. While color is predicted by a decoder (Eq. 8), there is no discussion of modeling dynamic texture (e.g., time-varying BRDFs, wrinkle maps, or view-dependent shading effects). The dynamic appearance is a result of the dynamic geometry, but the title suggests the texture itself is being modeled dynamically, which does not appear to be the case.

**Questions:**

1. The authors describe down-sampling N Gaussians to M graph nodes for the GNN. How are the deformations computed on these M nodes transferred back to the full set of N Gaussians for rendering?
2. Could the authors clarify the "Dynamic Texture" aspect of the title?
3. Section 4.3 and Table 2 state the best velocity window is T_v = 15, but Appendix D.1 says T_v = 11 yielded the highest performance. Could the authors clarify the difference in interpretation of these in selecting the final model?

---

> ### Author Response · Authors · 2025-11-27
>
> Many thanks to your valuable comments and questions, which help us a lot to improve our work. We address your questions as follows.
>
> ---
>
> ### **W1&Q1. On Gaussian primitives and graph nodes**
>
> Thank you for pointing out this important clarification. In our pipeline, the deformation predicted on the M graph nodes directly determines the final set of Gaussians used for rendering. Specifically, the system does not propagate deformations from M nodes back to the original N canonical Gaussians. Instead, the down-sampling step is a structural filtering process that reduces the initial N Gaussians to M Gaussians, which then serve as the complete set of Gaussian primitives throughout training and rendering.
>
> The motivation for this design is twofold: (1) to prevent overfitting when learning second-order, autoregressive dynamics, and (2) to keep the dynamical model stable and computationally tractable. As shown in our capacity analysis (Table 2; SMAD capacity) , we determined that M = 40K Gaussians provides the best trade-off between fidelity and stability.
>
> Thus, there is no need for an additional interpolation or skinning scheme (e.g., nearest-node rigidity or LBS-like blending) between M nodes and N Gaussians—the graph nodes are the final Gaussians. All deformations predicted by the GNN are applied directly to these M Gaussian primitives, which are then rendered without further upsampling. For clarity, we update Sec.3.1 in the revised manuscript.
>
> ---
>
> ### **W2&Q2. On Title "Dynamic Texture Modeling"**
> We appreciate the reviewer’s careful reading and fully agree with the concern regarding the wording in the current title. Our method’s primary technical novelty lies in dynamic geometry and motion modeling—i.e., the autoregressive deformation of Gaussian primitives (position, covariance)—rather than explicit modeling of time-varying texture properties such as BRDFs, wrinkle maps, or view-dependent shading.
>
> We will revise the title to more accurately reflect the methodological focus on structural and motion dynamics rather than explicit texture modeling.
>
> A possible revised title is: “Dynamic Appearance Modeling of 3D Clothed Gaussian Avatars from a Single Video” or, alternatively: “Secondary Motion-Aware Deformation of 3D Clothed Gaussian Avatars from a Single Video”.
>
> We will adopt one of these clearer alternatives in the camera-ready version to avoid overstating the role of texture and to better align the title with the technical contributions detailed throughout the paper.
>
> ---
>
> ### **Q3. Typo of velocity window size**
>
> Thank you for pointing out this inconsistency. The mismatch between Table 2 and Appendix D.1 is due to a typo in Table 2. The correct configuration for the velocity-encoding window is as follows: B2: $\tau_{v} = 7$ , B3: $\tau_{v} = 15$, and B4: $\tau_{v} = 11$ (Ours). In other words, $\tau_{v} = 11$ is indeed the setting that yielded the best performance and the one used in our final model, fully consistent with Appendix D.1.
>
> The mistaken entries in Table 2 were inadvertently swapped during table formatting. We correct the Table in the revised version. This correction aligns all sections of the paper and resolves the confusion about how the final τ_v was selected.
>
> We appreciate the reviewer for catching this and the corrected values are reflected clearly in Sec.4.2 and Table 2. In addition, to avoid confusion, we removed the words in Appendix D.1 that provided redundant information.
>
> ---
>
> ### **Suggestions**
> Regarding the reviewer’s suggestion to incorporate additional related works [1–7], we fully agree that these references are highly relevant to the broader discussion of loose clothing modeling, both in template-based and template-free settings. We sincerely thank the reviewer for the insightful and encouraging comments. We have carefully reviewed all the suggested papers and have updated the section of Related Work to properly situate our contributions within this line of research.
>
> ---

---

### Author Response · Authors · 2025-11-27

We sincerely thank all reviewers, **Wxo4, 1tPA, Gn6q, and ZfFP,** for their thoughtful and constructive evaluations of our work. We carefully addressed each concern in the individual responses; here, we provide a brief summary highlighting the core strengths acknowledged by the reviewers and clarifications for the main concerns.

---

### **Strengths Highlighted by Reviewers**

**Clear problem motivation and analysis.**

Multiple reviewers (e.g., Wxo4, ZfFP, Gn6q) appreciated our clear identification of the fundamental limitations of current monocular 3DGS avatars—template mismatch and lack of temporal modeling—and noted that the paper articulates these issues convincingly.

**Effectiveness of the proposed SMAD framework.**

Reviewers recognized the technical importance of adopting a second-order autoregressive formulation and velocity encoding. Wxo4 and ZfFP noted that our approach leads to substantial improvements in secondary motion and reduces artifacts such as skirt-splitting and needle artifacts. Ablation results were also acknowledged as meaningful.

**Impactful design choice: template-free, personalized initialization.**

The personalized Gaussian initialization (PGI) stage was viewed as a strong and principled alternative to SMPL-based pipelines, addressing the long-standing issue of naked-body bias when modeling loose garments (Wxo4, ZfFP).

**Contribution of the LoCo-Human dataset.**

Reviewers noted that our in-the-wild dataset fills an important gap by providing loose-clothing and dynamic motions not available in existing benchmarks (Wxo4, Gn6q).

---

### **Clarifications on Raised Concerns**

**Ablation on GNN vs. MLP deformers (ZfFP).**

Following the reviewer’s suggestion, we added a controlled experiment comparing our GNN-based deformer with a strong MLP baseline inspired by prior secondary-motion emulators. The GNN showed superior robustness on unseen motions.

**Training cost and PGI trade-off (ZfFP).**

We added a detailed analysis showing how performance scales with training time. Unlike prior methods, our model is improved with longer training, demonstrating higher effective capacity for dynamic appearance modeling.

**Physics-inspired vs. physically consistent formulation (Gn6q).**

We clarified that our method is *physics-inspired*, not a physically calibrated simulator. The GNN implicitly learns interaction patterns without enforcing explicit ground-truth colliders and 4D cloth and body scans. We revised the text to avoid overstating physical grounding.

**Graph design & relation to prior mesh-based dynamics (Gn6q).**

We clarified the differences between prior ground-truth clothed template mesh-based graph methods and our template-free Gaussian graph built directly on canonical clothed Gaussians under monocular constraints, an unexplored combination in prior 3DGS-based avatars.

**Generalization (1tPA).**

We performed extensive quantitative analyses—including motion-embedding similarity, t-SNE evaluation, and train/test/OOD comparisons with paired t-tests—which collectively show no significant performance drop on unseen motions. These results confirm that our autoregressive, velocity-based design provides stable generalization beyond the training distribution.

**Single-video setting vs. ActorsHQ/Goliath (1tPA).**

Our method is explicitly designed for monocular single-video input, whereas ActorsHQ and Goliath are designed for multi-view capture, tight clothing, or template-driven geometry—thus inconsistent with our problem setting.

---

We thank all reviewers again for their detailed feedback, which we believe has helped us further strengthen the clarity and presentation of our contributions. We have incorporated all clarifications and corrections, **highlighted in blue**, into the **revised manuscript**, which is included alongside this rebuttal.

---

### Meta-Review · Area_Chair_ikRQ · 2026-01-05

**Summary:**

This paper receives all positive ratings (8, 6, 6, 6). And reviewers' major concerns can be summarized into following points:
1) limited novelty (reviewer 1tPA, reviewer Gn6q, reviewer ZfFP).
2) unclear training, method and experiment details (reviewer Wxo4, reviewer ZfFP).
3) lack of some experiments (reviewer 1tPA, reviewer ZfFP).
4) method's limitation (person-specific (reviewer 1tPA), no physical consistency (reviewer 1tPA, reviewer Gn6q)).

During rebuttal and discussion period, authors have provided responses and well addressed concerns in 2), 3) and 4). In terms of 1), AC agrees with reviewers that the novelty is limited, since of GS+LBS is limited, and 3D-GS simulator (GNN-based or not) has been introduced in previous literature [1-5].

- [1] PGC: Physics-Based Gaussian Cloth from a Single Pose, https://phys-gaussian-cloth.github.io/
- [2] GausSim: Foreseeing Reality by Gaussian Simulator for Elastic Objects, https://arxiv.org/abs/2412.17804
- [3] MPMAvatar: Learning 3D Gaussian Avatars with Accurate and Robust Physics-Based Dynamics, https://arxiv.org/abs/2510.01619
- [4] Learning 3D-Gaussian Simulators from RGB Videos, https://arxiv.org/abs/2503.24009
- [5] Gaussian Garments: Reconstructing Simulation-Ready Clothing with Photo-Realistic Appearance from Multi-View Video, https://eth-ait.github.io/Gaussian-Garments/

Despite of this, overall this submission presents an effective method with clear motivation. Thus the decision is accept.

**Reviewer Concerns:**

Concerns are:
1) limited novelty (reviewer 1tPA, reviewer Gn6q, reviewer ZfFP).
2) unclear training, method and experiment details (reviewer Wxo4, reviewer ZfFP).
3) lack of some experiments (reviewer 1tPA, reviewer ZfFP).
4) method's limitation (person-specific (reviewer 1tPA), no physical consistency (reviewer 1tPA, reviewer Gn6q)).

AC found 2), 3), 4) have been addressed, and 1) is still outstanding.

**Reviewer Scores:**

AC think reviewers would stay positive

---

### Decision · Program_Chairs · 2026-01-26

Accept (Poster)